# Learning Unified Representation of 3D Gaussian Splatting

**Yuelin Xin**[1]  **Yuheng Liu**[1]  **Xiaohui Xie**[1]  **Xinke Li**[2,†]

[1]UC Irvine    [2]City University of Hong Kong

## Abstract

A well-designed vectorized representation is crucial for the learning systems natively based on 3D Gaussian Splatting. While 3DGS enables efficient and explicit 3D reconstruction, its parameter-based representation remains hard to learn as features, especially for neural-network-based models. Directly feeding raw Gaussian parameters into learning frameworks fails to address the non-unique and heterogeneous nature of the Gaussian parameterization, yielding highly data-dependent models. This challenge motivates us to explore a more principled approach to represent 3D Gaussian Splatting in neural networks that preserves the underlying color and geometric structure while enforcing unique mapping and channel homogeneity. In this paper, we propose an embedding representation of 3DGS based on continuous submanifold fields that encapsulate the intrinsic information of Gaussian primitives, thereby benefiting the learning of 3DGS. Implementation available at `https://github.com/cilix-ai/gs-embedding`.

## 1 Introduction

Recent advances in 3D Gaussian Splatting (3DGS) (Kerbl et al., 2023) have established it as a powerful technique for representing and rendering 3D scenes, enabling high-fidelity, real-time novel view synthesis through explicit parameterization of Gaussian primitives. This representation has catalyzed a growing body of work exploring learning-based methods that operate directly on Gaussian primitives, supporting tasks such as compression (Shin et al., 2025), generation (Yi et al., 2024; Xie et al., 2025), and understanding (Guo et al., 2024). In these pipelines, the native parameterization $\theta = \{\boldsymbol{\mu}, \mathbf{q}, \mathbf{s}, \mathbf{c}, o\}$ is often adopted as the input or output of neural architectures.

Despite its effectiveness in optimization-based reconstruction, we identify fundamental limitations when this parametric representation is employed as a learning space for neural networks. Specifically, the native parameterization $\theta$ conflicts with the inductive biases of standard neural architectures in three critical ways. First, the mapping from parameters to rendered output exhibits **non-uniqueness**. Ambiguities such as quaternion sign duality and symmetry-induced variances create a one-to-many mapping. This mapping creates "embedding collisions" where distinct parameter inputs producing identical visual outputs generate conflicting supervision signals, leading to training instability and poor convergence (Bengio et al., 2013; Wang & Isola, 2020). Second, the parameter components suffer from **numerical heterogeneity**. Spatial positions span large magnitudes while quaternions remain unit-normalized, violating the homogeneous feature distribution assumption required for effective gradient flow (Ioffe & Szegedy, 2015). Third, these parameters inherently reside on **distinct mathematical manifolds**, such as positions in $\mathbb{R}^3$, rotations in $SO(3)$, and appearance in spherical harmonic coefficients. Forcing these non-Euclidean variables into the Euclidean feature spaces of standard encoders breaks their intrinsic geometric structure, making the representation difficult to compress or regularize effectively.

These theoretical misalignments translate into substantial practical failures across various applications. Our empirical analysis reveals that parametric encoders are critically unstable; for instance, due to the sign ambiguity, simply negating a quaternion (a mathematically equivalent rotation) can cause complete decoding failure in parameter-trained autoencoders, see App. Fig. b. This instability extends to generative tasks involving latent manipulation, where we observe that linear interpolation

---

† Corresponding author.

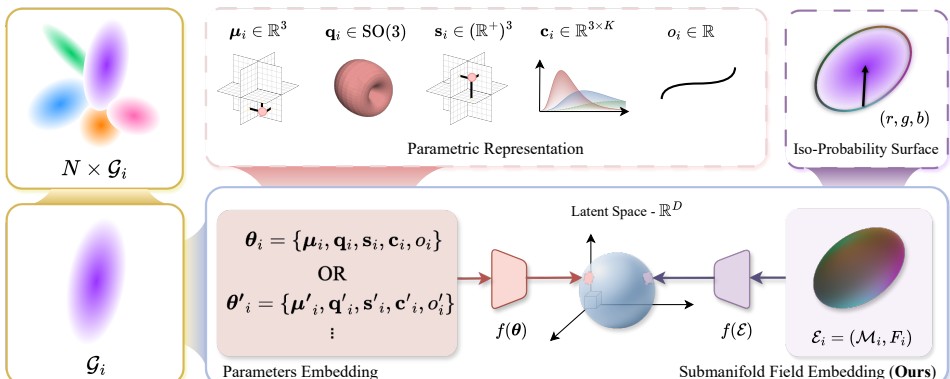

Figure 1: A scene of $N$ Gaussian primitives can be represented by $N$ sets of parameters $\boldsymbol{\theta}$ (shown in pink). Data in this parametric space resides on different manifolds and is heterogeneous and non-Euclidean, introducing challenges for encoders to fit disparate data manifolds implicitly. Shown in purple is the proposed representation, instead of relying on Gaussian parameterization, we introduce a canonical submanifold field space $(\mathcal{M}, F)$ that uniquely represents a Gaussian primitive with an iso-probability surface.

in the parametric latent space results in geometric "jitters" and unnatural transitions. Furthermore, parametric embeddings lack robustness to noise, with minor perturbations causing disproportionate reconstruction errors. Crucially, for downstream tasks extensively explored in recent work including generative modeling (Yi et al., 2024; Zhou et al., 2024b), compression (Shin et al., 2025; Girish et al., 2024), and editing (Chen et al., 2024a), these flaws manifest as discontinuous latent spaces and inefficient encoding. Instead of capturing robust geometric semantics, models are forced to resolve parameter ambiguities with more efforts, leading to sub-optimal performance in learning-based reconstruction (Charatan et al., 2024) and limited generalization across diverse domains.

We thus propose a principled alternative that represents each Gaussian primitive as a continuous field defined on its iso-probability surface. This submanifold field representation establishes a unique correspondence between Gaussians and their geometric-photometric properties, removing the ambiguities of parametric representations. By discretizing this field as a colored point cloud sampled from the probability surface, we obtain a numerically uniform and geometrically consistent representation. We further employ a variational autoencoder to learn embeddings from these discretized submanifold fields, together with a Manifold Distance metric based on optimal transport that better correlates with perceptual quality than parameter-space distances. Extensive experiments show higher reconstruction quality, stronger cross-domain generalization, and more robust latent representations. The learned embedding space demonstrates potential for downstream applications through unsupervised segmentation and neural field decoding with the proposed embeddings. To summarize the contribution of this work, we:

- identify and formally characterize the fundamental limitations of parametric Gaussian representations for neural learning, including non-uniqueness and numerical heterogeneity.

- propose a submanifold field representation that provides provably unique and geometrically consistent encoding of Gaussian primitives.

- develop a variational autoencoder framework incorporating a novel Manifold Distance metric based on optimal transport theory for effective learning in the submanifold field representation space with extensive experimental evidence.

## 2 RELATED WORKS

**3D Gaussian Splatting.** Since its re-introduction by Kerbl et al. (2023), 3DGS has rapidly become a core method for novel view synthesis and 3D representation. By placing explicit Gaussian primitives in 3D space and employing efficient rasterization and accumulation, 3DGS achieves real-time rendering with high fidelity (Bao et al., 2025; Lin et al., 2025c). Several studies improve efficiency (Jo et al., 2024; Lee et al., 2024), while others leverage large-scale datasets for generalization (Ma et al.,

2025; Li et al., 2025a). At the application level, 3DGS has been adopted for digital human (Li et al., 2024; Kocabas et al., 2024; Wang et al., 2025), self-driving scene modeling (Zhou et al., 2024a;c; Yan et al., 2023), and physics-based simulation (Jiang et al., 2024; Xie et al., 2024; Zhong et al., 2024). Beyond fixed Gaussian parameters, another line augments primitives with latent embeddings to capture semantics, open-vocabulary understanding (Qin et al., 2024) and deformation modeling (Zhobro et al., 2025). These efforts demonstrate that 3DGS provides high-fidelity appearance and serves as a versatile representation with broad potential (Sun et al., 2025).

**3DGS Parameters Regression.** To enable fast and flexible reconstruction without per-scene optimization, recent works seek to directly obtain Gaussian splats through feedforward prediction networks. For example, Charatan et al. (2024); Chen et al. (2024b) proposed to predict Gaussian parameters directly from multi-view input, while Zheng et al. (2024) generate pixel-wise parameter maps and lift them to 3D via depth estimation. This paradigm has been extended to pose-free settings (Hong et al., 2024; Chen et al., 2024c; Tian et al., 2025), and transformer-based methods further improve scalability and generalization (Li et al., 2025b; Jiang et al., 2025; Lin et al., 2025b). Overall, these methods directly output Gaussian parameters from neural networks, showing that 3DGS can serve as an effective target for network-based prediction in efficient reconstruction.

**Embedding Gaussian Primitives.** Recent works move beyond reconstruction and encode Gaussian parameters into latent spaces for tasks such as generation, editing, and compression. Zhou et al. (2024b); Lin et al. (2025a); Wewer et al. (2024) learn structured latent variables from 3D Gaussian space to fulfill generation tasks. Editing and style-transfer methods use diffusion or style conditioning to manipulate Gaussians primitives in latent or rendering spaces (Chen et al., 2024a; Vachha & Haque, 2024; Lee et al., 2025; Palandra et al., 2024; Zhang et al., 2024; Kovács et al., 2024; Yu et al., 2024). Other works improve rendering quality by optimizing Gaussian parameters under diffusion priors (Tang et al., 2023; Yi et al., 2024; Chen et al., 2024d), while compression methods (Girish et al., 2024; Yang et al., 2025) reduce storage and computation by quantizing and embedding Gaussian parameters. These approaches show the potential of embedding Gaussians into neural latent spaces, but they assume Gaussian parameters are naturally compatible with neural learning, overlooking that these parameters were designed for optimization-based reconstruction. This oversight underlies our analysis in Section 3 and our proposal of a more suitable formulation.

## 3 METHOD

### 3.1 PRELIMINARIES: GAUSSIAN SPLATTING PARAMETERIZATION

A scene under 3D Gaussian Splatting is represented as a set of $N$ oriented, and view-dependently colored Gaussian primitives $\{\mathcal{G}_i\}_{i=1}^N$, each contributing to the final rendered image via rasterization and alpha compositing. Each Gaussian primitive $\mathcal{G}_i$ is usually represented by a parameter tuple $\boldsymbol{\theta}_i = \{\boldsymbol{\mu}_i, \mathbf{q}_i, \mathbf{s}_i, \mathbf{c}_i, o_i\}$, where:

- $\boldsymbol{\mu}_i \in \mathbb{R}^3$: the center position of the Gaussian in world coordinates;
- $\mathbf{q}_i \in \mathrm{SO}(3)$: a unit quaternion representing the local rotation;
- $\mathbf{s}_i \in (\mathbb{R}^+)^3$: scale parameters along the rotated axes;
- $\mathbf{c}_i \in \mathbb{R}^{3 \times K}$: spherical harmonic (SH) coefficients for view-dependent color for $K \in \mathbb{Z}$;
- $o_i \in \mathbb{R}$: a logit-transformed opacity value $\alpha_i = \sigma(o_i)$, where $\sigma$ is a sigmoid function.

The local geometry of the Gaussian is governed by its covariance matrix, constructed as

$$\Sigma_i = R(\mathbf{q}_i) \operatorname{diag}(\mathbf{s}_i)^2 R(\mathbf{q}_i)^\top, \tag{1}$$

where $R(\mathbf{q}_i)$ is the rotation matrix corresponding to the quaternion $\mathbf{q}_i$. This defines an ellipsoidal spatial density, centered at $\boldsymbol{\mu}_i$, whose shape and orientation determine the contribution of $\mathcal{G}_i$ to the rendered scene. The color at a given view direction $\mathbf{d} \in \mathbb{S}^2$ is computed per channel using SH basis functions denoted by

$$\mathrm{Color}_i(\mathbf{d}) = \left[ \mathrm{SH}_i^r(\mathbf{d}), \mathrm{SH}_i^g(\mathbf{d}), \mathrm{SH}_i^b(\mathbf{d}) \right]^\top, \tag{2}$$

where $\mathrm{SH}_i^c(\mathbf{d})$ in $c$−channel is calculated by $\sum_{l=0}^{L_{\max}} \sum_{m=-l}^{l} (\mathbf{c}_i)_{c,(l,m)} \cdot Y_l^m(\mathbf{d}) \, Y_l^m$ is the real-valued spherical harmonic of degree $l$ and order $m$. The final rendering aggregates contributions

from all $\mathcal{G}_i$ via a soft visibility-weighted compositing process. This native parameterization is well-suited for gradient-based scene optimization. However, it introduces significant challenges when used as a representation for learning.

## 3.2 PARAMETERIZATION IS ILL-SUITED AS A LEARNING SPACE

The parameter representation $\boldsymbol{\theta}$ poses fundamental challenges when used as a learning space for neural networks. We identify two critical issues: representation non-uniqueness and numerical heterogeneity. Each undermines the stability and effectiveness of neural network training.

**Representation Non-uniqueness.** The parametric representation suffers from a many-to-one mapping that violates basic requirements for stable learning. To understand this, we first formalize what rendering effect a single Gaussian primitive produces.

**Definition 1 (Single Gaussian Radiance Field (SGRF))** *A SGRF is a radiance field $\phi : \mathbb{R}^3 \times \mathbb{S}^2 \to \mathbb{R}^3$, The field is defined by the local density at point $\mathbf{x} \in \mathbb{R}^3$ along direction $\mathbf{d} \in \mathbb{S}^2$:*

$$\phi_{\mathcal{G}}(\mathbf{x}, \mathbf{d}) = \rho_{\mathcal{G}}(\mathbf{x}) \cdot c_{\mathcal{G}}(\mathbf{d}), \tag{3}$$

*where $\rho_{\mathcal{G}}(\mathbf{x}) = \exp\left(-\frac{1}{2}(\mathbf{x} - \boldsymbol{\mu})^\top \Sigma^{-1}(\mathbf{x} - \boldsymbol{\mu})\right)$ is a volume density function and $c_{\mathcal{G}}(\mathbf{d})$ is a color radiance field coupled with opacity. Specifically, given a parameter set $\boldsymbol{\theta} = \{\boldsymbol{\mu}, \mathbf{q}, \mathbf{s}, \mathbf{c}, o\}$, $\Sigma$ can be derived by equation 1 and $c_{\mathcal{G}}(\mathbf{d}) = \sigma(o) \cdot \mathrm{Color}(\mathbf{d})$ can be derived by equation 2.*

The SGRF, derived from the multi-Gaussian rendering framework by Kerbl et al. (2023), specifies how the final value at any pixel is rendered in a scene containing only one Gaussian splat. Furthermore, let $\Phi$ be the space of SGRFs, and $\Theta \subseteq \mathbb{R}^{|\boldsymbol{\theta}|}$ be the paramater space of Gaussian primitives, each parameter set $\boldsymbol{\theta} \in \Theta$ provides a complete representation that generates a correponding field $\phi_{\mathcal{G}} \in \Phi$. We indicate that a single SGRF may correspond to multiple parameterizations of Gaussian primitives, as formalized in the following proposition.

**Proposition 1 (Non-uniqueness of the SGRF Parametric Representation)** *The parametric representation of a SGRF is not unique. Formally, there exist at least two distinct parameter sets, $\boldsymbol{\theta}_1 \in \Theta$ and $\boldsymbol{\theta}_2 \in \Theta$ with $\boldsymbol{\theta}_1 \neq \boldsymbol{\theta}_2$, that generate the exact same field $\phi_{\mathcal{G}} \in \Phi$.*

The non-uniqueness is from quaternion sign ambiguity, geometric symmetries, and rotation-spherical harmonic interactions producing equivalent parameter combinations (see App. A for proof). The non-uniqueness of $\boldsymbol{\theta}$ will create "embedding collisions" where different parameter vectors produce identical rendered output (Wang & Isola, 2020). This makes the learning objective $\|\boldsymbol{\theta}_{\mathrm{pred}} - \boldsymbol{\theta}_{\mathrm{target}}\|_p$ ambiguous, as multiple parameter configurations can achieve the same visual result. The resulting conflicting gradients lead to training instability and poor convergence indicated by Bengio et al. (2013).

**Numerical Heterogeneity** The parameter components violate the homogeneous distribution assumption of standard neural architectures. Neural networks typically assume features share similar statistical properties for effective gradient flow (Ioffe & Szegedy, 2015). However, 3D Gaussian parameters span vastly different ranges. For example, pre-activation scales can range from $-15$ to $3$, while quaternions stay unit-normalized. More fundamentally, these parameters follow different distributions and live on different manifolds: positions $\boldsymbol{\mu} \in \mathbb{R}^3$, rotations $\mathbf{q} \in \mathrm{SO}(3)$, scales $\mathbf{s} \in (\mathbb{R}^+)^3$, and SH coefficients $c$ with exponential decay. Concatenating them ignores their heterogeneous nature. Small noises in quaternions can drastically alter geometry, while small noise in high-order SH coefficients is negligible, yet the network treats all dimensions equally.

The non-uniqueness and numerical heterogeneity of the native parameter space $\boldsymbol{\theta}$ make it unsuitable for neural network learning, which would generate unstable embeddings (see our experiments in Sec. 4.3 and App. D). We therefore introduce a submanifold field representation that ensures unique mappings and respects the geometric structure of 3D Gaussians.

## 3.3 REPRESENTATION ON SUBMANIFOLD FIELD

To address this issue, we propose converting each Gaussian primitive $\mathcal{G}_i$ to a novel geometric representation $\mathcal{E}_i$, which is a color field defined on a 2D submanifold in 3D Euclidean space, as illustrated

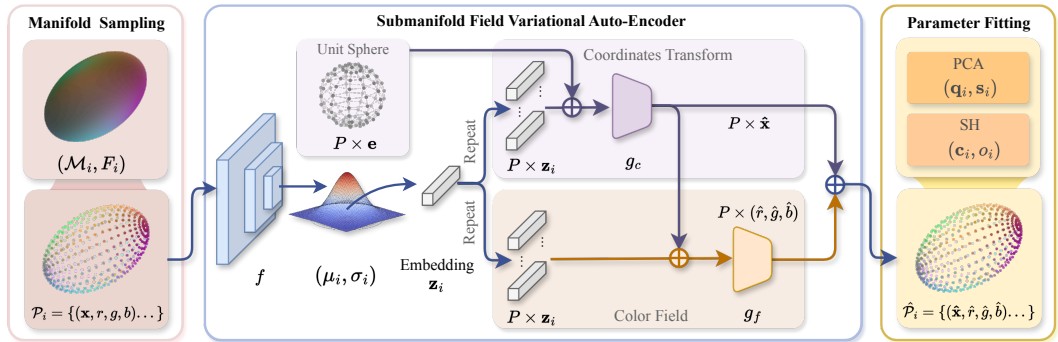

Figure 2: To embed the proposed submanifold field representation into a vector form suitable for neural networks, we devise a Submanifold Field Variational Auto-encoder (SF-VAE) that embeds any input submanifold field as a 32-D vector, then reconstructs the original parameter set $\boldsymbol{\theta}_i$. SF-VAE learns in our new representation space instead of the parametric space.

in Fig. 1. For a Gaussian density $\mathcal{N}(\mathbf{x}; \boldsymbol{\mu}_i, \Sigma_i)$, we define the iso-probability surface at fixed radius $r$ as:

$$\mathcal{M}_i = \left\{ \mathbf{x} \in \mathbb{R}^3 \mid (\mathbf{x} - \boldsymbol{\mu}_i)^\top \Sigma_i^{-1} (\mathbf{x} - \boldsymbol{\mu}_i) = r^2 \right\}, \tag{4}$$

which forms an ellipsoid surface, namely, a two-dimensional submanifold, centered at $\boldsymbol{\mu}_i$. On this submanifold, we define a field function:

$$F_i(\mathbf{x}) = \sigma(o_i) \cdot \text{Color}_i(\mathbf{d}_\mathbf{x}), \tag{5}$$

where $\mathbf{d}_\mathbf{x} = (\mathbf{x} - \boldsymbol{\mu}_i)/\|\mathbf{x} - \boldsymbol{\mu}_i\|$ denotes the unit direction vector for $\mathbf{x} \in \mathcal{M}_i$, and $\text{Color}_i(\cdot)$ represents the view-dependent color parameterization as in equation 2. Let $\mathbb{M}$ be the space of all possible iso-probability submanifolds as defined in equation 4, we define our unified representation space as:

$$\mathscr{E} = \left\{ \mathcal{E}_i = (\mathcal{M}_i, F_i) \mid \mathcal{M}_i \in \mathbb{M}, F_i : \mathcal{M}_i \to \mathbb{R}^3 \right\}, \tag{6}$$

The representation $\mathcal{E}_i \in \mathscr{E}$ encodes both geometric properties (shape, orientation) via $\mathcal{M}_i$ and appearance attributes (view-dependent color) via $F_i$ in a continuous framework. We have the following proposition (proof is provided in App. B).

**Proposition 2 (Uniqueness of Submanifold Field Representation)** *For every SGRF $\phi_\mathcal{G} \in \Phi$, there exists a unique corresponding representation $\mathcal{E} \in \mathscr{E}$. This establishes a one-to-one correspondence between the elements of $\Phi$ and $\mathscr{E}$. Formally, for any two distinct fields $\phi_{\mathcal{G},1}, \phi_{\mathcal{G},2} \in \Phi$, their corresponding representations $\mathcal{E}_1, \mathcal{E}_2 \in \mathscr{E}$ are also distinct.*

The submanifold field $\mathcal{E}_i$ thus provides a numerically stable and provably unique representation space on which we can safely build learning objectives and neural architectures.

### 3.4 ENCODE SUBMANIFOLD FIELDS AS EMBEDDINGS

We design a variational auto-encoder to encode submanifold field representation, shown in Fig. 2. The network architecture, learning objectives and dataset are introduced.

**Encoder-decoder Architecture.** We employ a point-cloud-based network to encode and decode one sub-manifold field. Particularly, we uniformly sample $P$ points from the submanifold field $\mathcal{E}$ as a colored point cloud $\mathcal{P} = \left\{ (\mathbf{x}_m, F(\mathbf{d}_{\mathbf{x}_m})) \right\}_{m=1}^P$. We then employ a PointNet (Qi et al., 2017) encoder $f$ to obtain latent embedding by $\mathbf{z} \sim f(\mathbf{z} \mid \mathcal{P})$ where $\mathbf{z} \in \mathbb{R}^D$ is the embedding with dimension $D$. The decoder $g$ consists of two neural networks, namely, the coordinates transform network $g_c : \mathbb{R}^3 \times \mathbb{R}^D \to \mathbb{R}^3$ and color field $g_f : \mathbb{R}^3 \times \mathbb{R}^D \to \mathbb{R}^3$. The decoded point cloud from decoder is given by,

$$\hat{\mathcal{P}} = g(\mathbf{z}, \mathcal{U}_{P'}) = \{g_c([\mathbf{e}_n, \mathbf{z}]), g_f([g_c([\mathbf{e}_n, \mathbf{z}]), \mathbf{z}])\}_{n=1}^{P'} \tag{7}$$

where $\mathcal{U}_{P'} = \{\mathbf{e}_n\}_{n=1}^{P'}$ is a set of coordinates sampled from a unit sphere surface. Such canonical set works as the initial input for two implicit functions $g_c$ and $g_f$, and queries new coordinates and

color field. Furthermore, to recover the original Gaussian parameters $\boldsymbol{\theta}_i$ for rendering purposes, we estimate the covariance matrix $\Sigma_i$ by principal component analysis (PCA), and SH coefficients $\mathbf{c}_i$ by fitting the spherical harmonics to $\hat{\mathcal{P}}$.

**Learning Objectives.** We introduce *Manifold Distance* (M-Dist) for the reconstruction objective in encoder-decoder training. Given two submanifold fields $\mathcal{E} = (\mathcal{M}, F)$ and $\hat{\mathcal{E}} = (\hat{\mathcal{M}}, \hat{F})$, we propose to measure their similarity based on the Wasserstein-2 distance from optimal transport defined as

$$W_2^2(\mathcal{E}, \hat{\mathcal{E}}) = \inf_{\gamma \in \Gamma(\hat{\sigma}, \hat{\sigma}')} \int_{\mathcal{M} \times \hat{\mathcal{M}}} d^2\big((\mathbf{x}, c_x), (\mathbf{y}, c_y)\big) d\gamma(\mathbf{x}, \mathbf{y}) \tag{8}$$

where $c_x = F(\mathbf{d_x})$, $c_y = \hat{F}(\mathbf{d_y})$, $\Gamma(\hat{\sigma}_i, \hat{\sigma}_j)$ is the set of all joint probability measures (transport plans) with marginals $\hat{\sigma}_i$ and $\hat{\sigma}_j$, and the ground distance is defined as

$$d^2\big((\mathbf{x}, c_x), (\mathbf{y}, c_y)\big) = \|\mathbf{x} - \mathbf{y}\|_2^2 + \lambda \|c_x - c_y\|_2^2, \tag{9}$$

with $\lambda \in \mathbb{R}^+$ balancing spatial and color terms. In practice, both $\mathcal{M}$ and $\hat{\mathcal{M}}$ are discretized as colored point clouds $\mathcal{P}$ and $\hat{\mathcal{P}}$. The empirical Wasserstein-2 distance $\hat{W}$ is then computed between these point clouds by

$$\hat{W}_2^2(\mathcal{P}, \hat{\mathcal{P}}) = \min_{\mathbf{\Gamma} \in \Gamma(\hat{\sigma}, \hat{\sigma}')} \sum_{(\mathbf{x}_i, c_{x_i}) \in \mathcal{P}} \sum_{(\mathbf{y}_j, c_{y_j}) \in \hat{\mathcal{P}}} \mathbf{\Gamma}_{ij} \left( d^2\big((\mathbf{x}_i, c_{x_i}), (\mathbf{y}_j, c_{y_j})\big) \right). \tag{10}$$

Finally, the learning objective for variational auto-encoder is

$$\mathcal{L}_{\text{VAE}} = \mathbb{E}_{\hat{\mathcal{P}} \sim \text{VAE}(\mathcal{P})} \left( \hat{W}_2^2(\mathcal{P}, \hat{\mathcal{P}}) + \beta \cdot d_{\text{KL}} \left( f(\mathbf{z} \mid \mathcal{P}) \| \mathcal{N}(0, \mathbf{I}) \right) \right), \tag{11}$$

where $\text{VAE}(\mathcal{P}) = g(f(\mathbf{z} \mid \mathcal{P}), \mathcal{U}_{P'})$ and the second term is the KL divergence loss for variational auto-encoder implementation, and $\beta$ is a balance factor.

**Dataset Preparation.** Since this embedding model only encodes single Gaussian primitives, which have no semantic meaning out of a scene's global context, we can use a *randomly generated dataset* of Gaussian primitives to train this model, thus making it domain-invariant to data. The implementation details of this generated dataset can be found in App. C.

# 4 EXPERIMENTS

## 4.1 EVALUATION SETUP

**Baseline Implementation Details.** To isolate the effect of representation choice, we adopt a self-implemented encoder–decoder framework for both the parametric representation $\boldsymbol{\theta}$ and the proposed submanifold field representation $\mathcal{E}$. While comparisons with existing 3DGS learning methods are possible, they typically involve task-specific architectures that confound the role of representation itself. Direct reuse of prior pipelines would not yield a controlled comparison, so we implement both baselines in the same VAE-style framework to attribute differences solely to the representation.

We implement and train three size-matched embedding models: our submanifold field VAE (Sec. 3.4), and two baseline parametric VAEs operating directly on $\boldsymbol{\theta}$. For the parametric models, each Gaussian primitive is represented as a 56-D vector ($3+4+3K+1$ for $L_{\max}=3$), omitting global coordinates to match the SF-VAE setting. A three-layer MLP encodes this input to a 32-D latent ($56 \rightarrow 512 \rightarrow 512 \rightarrow 32 \times 2$), and the decoder, either uses a MLP to map the latent back to $\hat{\boldsymbol{\theta}}_i$, or uses the same decoder of SF-VAE to map to $\hat{\mathcal{P}}$. This setting further decouples evaluation results with the training objective functions. Apart from input dimension, all models share identical depth, width, latent size (32), and optimizer settings (all using Adam), ensuring a matched capacity.

**Datasets.** We evaluate the proposed representation and compare it with the baseline primarily using two datasets. For object-level tasks, we utilize ShapeSplat (Ma et al., 2025), a large-scale 3DGS dataset derived from ShapeNet (Chang et al., 2015), comprising 52K objects across 55 categories. For scene-level experiments, we employ Mip-NeRF 360 (Barron et al., 2022), which contains 7 medium-scale scenes with abundant high-frequency details. Additionally, unless stated otherwise,

Table 1: Reconstruction quality comparison for object-level (ShapeSplat) and scene-level (Mip-NeRF 360) datasets. All models trained on the randomly generated dataset. The three models have a parameter count of 0.62M, 0.66M and 0.62M respectively. The relatively extreme perceptual metrics values in ShapeSplat come from the use of background during measurement.

| Input Representation | Encoder | Decoder | PSNR ↑ | SSIM ↑ | LPIPS↓ | M-Dist | $L_1$-Dist |
|---|---|---|---|---|---|---|---|
| **ShapeSplat** | | | | | | | |
| Parametric | MLP | MLP | 37.512 | 0.888 | 0.152 | 0.184 | 0.040 |
| Parametric | MLP | SF-VAE | 44.725 | 0.896 | 0.136 | 0.051 | 0.097 |
| Submanifold Field | SF-VAE | SF-VAE | **63.408** | **0.990** | **0.010** | 0.041 | 0.098 |
| **Mip-NeRF 360** | | | | | | | |
| Parametric | MLP | MLP | 18.818 | 0.564 | 0.452 | 0.510 | 0.034 |
| Parametric | MLP | SF-VAE | 20.923 | 0.730 | 0.359 | 0.055 | 0.173 |
| Submanifold Field | SF-VAE | SF-VAE | **29.833** | **0.953** | **0.079** | 0.048 | 0.179 |

Table 2: Reconstruction quality comparison under cross-domain setting. All models trained on either ShapeSplat or Mip-NeRF 360 dataset are tested on another dataset. We show that the generalization ability of SF Embedding framework is inherently domain-agnostic even without random data.

| Train set | Test set | Input Represent. | Encoder / Decoder | PSNR ↑ | SSIM ↑ | LPIPS ↓ |
|---|---|---|---|---|---|---|
| ShapeSplat | Mip-NeRF. | Parametric | MLP / MLP | 9.753 | 0.356 | 0.615 |
| | | Parametric | MLP / SF-VAE | 14.845 | 0.675 | 0.336 |
| | | Submanifold Field | SF-VAE / SF-VAE | 19.194 | 0.821 | 0.309 |
| Mip-NeRF. | ShapeSplat | Parametric | MLP / MLP | 55.624 | 0.957 | 0.067 |
| | | Parametric | MLP / SF-VAE | 60.777 | 0.987 | 0.013 |
| | | Submanifold Field | SF-VAE / SF-VAE | 62.576 | 0.990 | 0.014 |

we train the embedding models using the randomly generated Gaussian primitive dataset, with 500K randomly generated data samples; implementation details are provided in App. C.

**Evaluation Metrics.** To comprehensively assess both perceptual fidelity and representation quality, we report PSNR, SSIM, and LPIPS on rasterized reconstructions against ground truth Gaussian splats, as well as $L_1$ distance in the Gaussian parameter space. Crucially, we also include our proposed Manifold Distance (M-Dist) as an evaluation criterion. By cross-comparing M-Dist with parameter-space distances ($L_1/L_2$), we can demonstrate that M-Dist aligns more closely with perceptual "gold standard" metrics such as PSNR and LPIPS, validating our claims in Sec. 3

### 4.2 EVALUATION ON REPRESENTATION LEARNING FRAMEWORK

**Zero-shot Reconstruction.** We present a comprehensive quantitative and qualitative analysis of reconstruction quality for both object-level and scene-level data, as summarized in Tab. 1 and Fig. 3. All models are trained on the same randomly generated 3D Gaussian primitives dataset and evaluated on ShapeSplat and Mip-NeRF 360, using three matched encoder-decoder configurations to control for bias. Across all perceptual metrics (PSNR, SSIM, LPIPS), the submanifold field representation consistently outperforms parametric baselines. For example, on ShapeSplat, SF-VAE achieves substantially higher PSNR and SSIM and a much lower LPIPS, indicating both improved fidelity and perceptual quality. Similar performance gains are observed in scene-level reconstruction, where the submanifold field model demonstrates better performance across diverse spatial contexts.

Importantly, the *Manifold Distance* (M-Dist) metric shows a stronger empirical correlation with quality metrics like PSNR and LPIPS than traditional $L_1$ parameter distances, supporting our claim that M-Dist is a more robust and meaningful similarity measure for 3D Gaussian representations, truthfully reflecting perceptual differences rather than merely parameter discrepancies. The consistent improvement margin across both datasets highlights the advantage of learning in the submanifold field space, which better preserves intrinsic structure and view-dependent appearance, confirming the efficacy of our representation for high-fidelity 3D Gaussian modeling.

**Cross-domain Reconstruction.** We also evaluate cross-domain generalization by training on one real-world dataset and testing on the other (object-level ↔ scene-level) under an identical training protocol and capacity budget as in the reconstruction study. Concretely, we train either the proposed

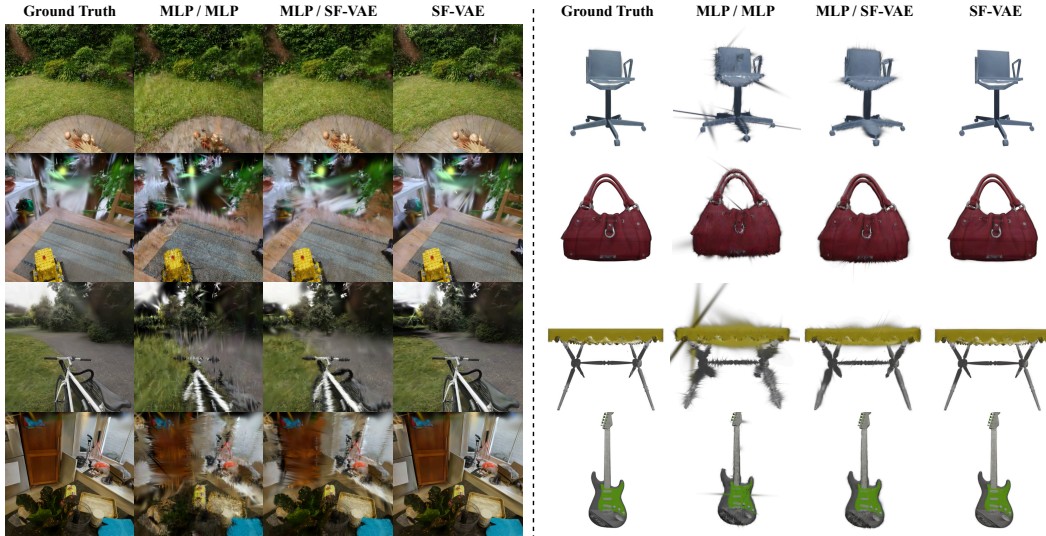

Figure 3: Qualitative results for rasterized reconstruction. Samples selected arbitrarily from Mip-NeRF 360 and ShapeSplat. Parametric models can induce confusion in parameter space, failing to embed and restore the correct Gaussian parameters.

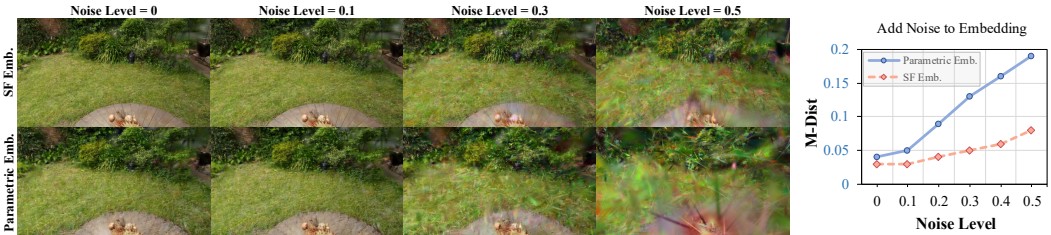

Figure 4: Reconstruction results using embeddings with noise. **Left:** Visualization of reconstructed scene from noisy embeddings of Gaussian parameters (MLP) and SF-VAE. **Right:** Comparison on M-Dist for different noise levels added to embedding space, tested on Mip-NeRF 360. *Noise level is defined as the ratio between the noise magnitude and the embedding variance.*

SF-VAE or the parametric MLP baselines on a source set and evaluate on a target set, rendering novel views and reporting PSNR, SSIM, and LPIPS averaged over test samples (see Tab. 2). Across both transfer directions, the SF-based embedding consistently achieves higher reconstruction quality than the parametric baseline, indicating reduced sensitivity to domain-specific statistics (e.g., scale, lighting, SH complexity). Particularly, comparing these transfer results with Tab. 1, we find that the model trained on synthetic random data actually outperforms the models transferred from real-world domains. This indicates that our random learning strategy effectively strips away domain priors, which establishes our approach as a unified representation, where a single model trained on synthetic data can fundamentally generalize to real-world contexts.

## 4.3 SENSITIVITY STUDY OF REPRESENTATION

**Robustness to Noise.** To evaluate the submanifold field embedding's robustness to noise, we gradually add higher levels of gaussian noise to the embedding space of the parametric model and the submanifold field model and test their reconstruction quality and M-Dist. To ensure fair comparison, we use noise level as a ratio to variance instead of absolute noise magnitude. As shown in Fig. 4, the embedding space of submanifold field model is more robust to random perturbation, this makes submanifold field embeddings a better learning target since it is less sensitive to potential noise introduced by downstream regression.

**Latent Space Interpolation.** To evaluate the regularity of the latent space of the proposed representation, we randomly sample pairs of source and target Gaussian primitives $\mathcal{G}_s$ and $\mathcal{G}_t$ and linearly interpolate each pair for a fixed number of steps $n = 7$. Compared with parametric space, the inter-

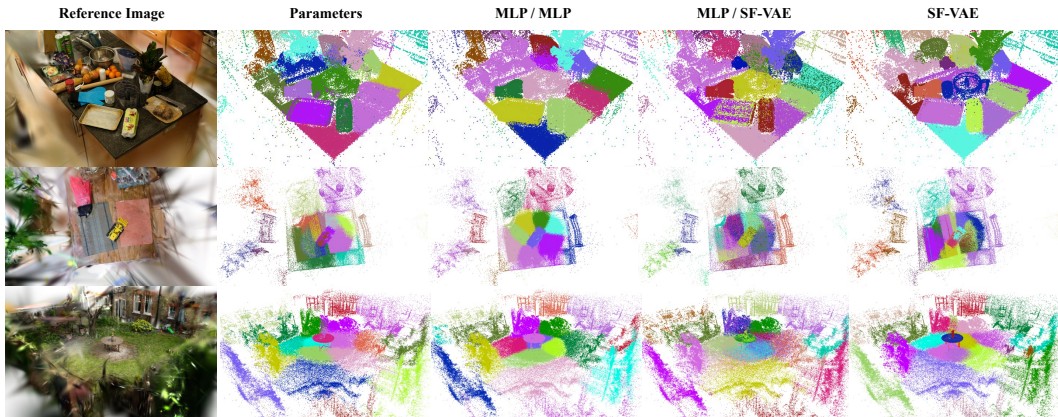

Figure 5: Unsupervised graph clustering based on raw Gaussian parameters and various embeddings. Submanifold field embeddings show better preservation of detailed semantics, showing its downstream applicability.

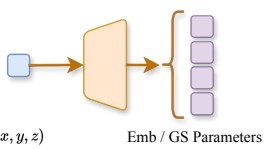

$(x, y, z)$     Emb / GS Parameters

Figure 6: Setting of a Gaussian Neural Field, we compare between the prediction target SF embedding and raw GS parameters.

Table 3: Comparison between Gaussian Neural Fields trained using submanifold field embeddings and raw Gaussian parameters. Top: ShapeSplat, bottom: Mip-NeRF 360.

| Target | PSNR ↑ | SSIM ↑ | LPIPS ↓ | # Params |
|---|---|---|---|---|
| Raw GS Parameter | 51.660 | 0.925 | 0.141 | 0.21M |
| SF Embedding | 58.619 | 0.980 | 0.043 | 0.20M |
| Raw GS Parameter | 19.922 | 0.648 | 0.410 | 1.87M |
| SF Embedding | 24.395 | 0.804 | 0.261 | 1.85M |

polation in submanifold field embedding space shows a smooth transition path, while interpolation in parametric space shows undesired jitter in rotation and scale, indicating space irregularities, see App. D. This highlights the motivation to learn in the unified submanifold field embedding space.

## 4.4 REPRESENTATION APPLICABILITY

**Unsupervised Clustering.** To further probe the semantic structure of the learned embedding spaces, we perform unsupervised graph clustering on both the raw Gaussian parameter space and the embedding outputs of each model. As visualized in Fig. 5, clusters formed in the submanifold field embedding space exhibit more detailed semantic separation against the reference images compared to those formed using normalized parameters or parametric embeddings. For example, SF-VAE's embedding clustering in the first line of Fig. 5 outlines clearer separation of foreground objects with the background. The clusters appear smoother, less noisy, and with clearer boundaries, showing an ability to distinguish between different entities. This indicates that the submanifold field embedding captures more dense semantics and discriminative features, validating its usefulness.

**Gaussian Neural Fields.** To validate the potential of our representation for advanced downstream tasks, we introduce the *Gaussian Neural Field* (GNF). Drawing inspiration from the decoding structures in generative diffusion models (e.g., DiffGS by Zhou et al. (2024b)) and neural compression frameworks (Wu & Tuytelaars, 2024), the GNF functions as a coordinate-based neural implicit field as illustrated in Fig. 6. Specifically, it employs a lightweight MLP (architecture detailed in App. D.4) to learn a continuous mapping from spatial coordinates $\mathbf{x}_i$ to per-primitive descriptors. This setup allows us to evaluate the "learnability" of our representation: while regressing heterogeneous raw parameters $\boldsymbol{\theta}_i$ often leads to optimization difficulties, our unified SF embeddings provide a smooth and well-conditioned target for the neural field. As evidenced in Tab. 3 and visualization in App. D.4, the SF-guided GNF outperforms the parameter-based baseline in visual fidelity with equivalent training effort. This indicates that our representation is more friendly to neural networks, hinting at its utility for potential downstream generative and compression tasks.

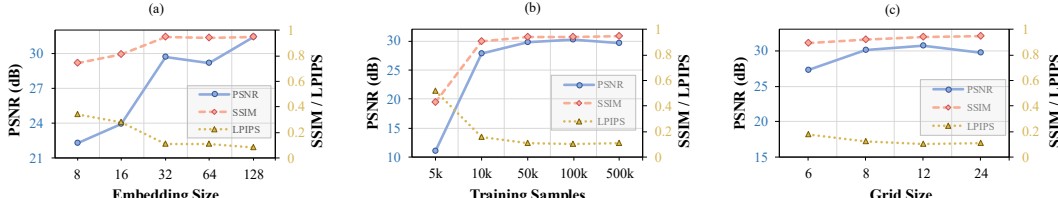

Figure 7: Behavior studies tested on Mip-NeRF 360. From left to right: (a) embedding dimension, (b) generated training dataset size, (c) Submanifold Field discretized (i.e., point sample) grid size.

### 4.5 MORE STUDIES ON IMPLEMENTATION DETAILS

**Ablation Study on SF-VAE Designs.** We provide performance comparison with different framework designs based on Mip-NeRF 360. (1) For encoder $f$, we tested DGCNN encoder (Wang et al., 2019) beyond our implementation, where the DGCNN encoder achieves comparable reconstruction fidelity while it is roughly $1.75\times$ slower in encoding and uses roughly $2\times$ more GPU memory during inference; (2) For the decoder's unit sphere grid, we tested a 2D grid implementation with matching grid size, 2D grid achieves similar reconstruction results but takes more iterations to converge; (3) We implemented two versions of the fitting module: a GPU-based version using FP32 with batching and Cholesky decomposition, and a CPU-based one using FP64 without batching and Least Squares solver. Experiments show that the GPU version introduces only negligible quality degradation (0.4 PSNR and 0.01 SSIM), while achieving an average speedup of $85\times$ with a batch size of 4096.

**Behavior Study of Latent Space Dimension.** We evaluated different embedding space dimensions for the SF-VAE model to meet the best trade-off between compression and reconstruction quality. All models are trained on the generated dataset with $L = 3$ order Spherical Harmonics. Results shown in Fig. 7 (a), 32 is the optimal balance point between reconstruction quality and latent space compression. All values are tested with baseline input/output of $P = 12^2$. While this work does not specifically focus on compression effectiveness, embedding space robustness shown in Sec. 4.3 suggests potential in further latent tokenization and quantization.

**Behavior Study of Training Set Size.** To determine the number of random training samples required to achieve the best reconstruction results, we vary the trainiing sample size from 5K to 500K (baseline), see Fig. 7 (b). The results indicate the proposed representation is data efficient. When only using 2% of the baseline training sample, our model can achieve close-to-baseline performance.

**Behavior Study of SF Discretized Size.** To ensure the submanifold fields are truthfully represented in a discrete manner, we evaluate different sample sizes $P$, see Fig. 7 (c). Going from the lowest tested $P = 6^2$ to the baseline $P = 12^2$, we observe a gradual improvement in reconstruction quality, while going above $P = 12^2$ yields very little improvement. Since $P$ directly correlates to the computational efficiency of the submanifold field model (see below), we keep $P = 12^2$.

**Computational Efficiency.** Increasing the point sample size $P$ increases computation and memory. Encoding time remains low since a lightweight PointNet-based encoder shares weights with all input points, giving an inference speed of 1.72s per 1 million Gaussians for $P = 12^2$ with a batch size of 4096 on an RTX 5090. Decoding time is 4.20s per 1 million Gaussians, from embedding to Gaussian parameters. The complexity is $O(P)$ or $O(n^2)$ w.r.t. the grid size $n$. We utilize the advantage of GPU parallel computation to boost the calculation for parameter fitting module. In the 4.20s of decoding time, the fitting module (PCA + SH fitting) consumes only about 0.48s which is negligible for large Gaussian scenes.

## 5 CONCLUSION AND LIMITATIONS

We introduced a geometry-aware *submanifold field* representation for 3D Gaussian Splatting that maps each primitive to a color field on an iso–probability ellipsoid and proved the mapping is injective over core attributes. Built on this representation, our SF–VAE learns semantically meaningful latents and yields higher-fidelity reconstructions and stronger zero-shot generalization than capacity-matched raw-parameter baselines. **Limitations and Outlook.** Our current setup operates at the single-Gaussian level, while this ensures data invariance, it omits explicit inter-splat structure modeling for more complex representation learning.

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
