# APPENDIX:
# LEARNING UNIFIED REPRESENTATION OF 3D GAUSSIAN SPLATTING

**Yuelin Xin**[*]**, Yuheng Liu & Xiaohui Xie**
Department of Computer Science
University of California, Irvine
{yuelix1,yuhenl22,xhx}@uci.edu

**Xinke Li**
Department of Data Science
City Univeristy of Hong Kong
xinkeli@cityu.edu.hk

## A    PROOF OF PROPOSITION 1

We construct two distinct parameter sets, $\boldsymbol{\theta}_1 = \{\mathbf{q}_1, \mathbf{s}_1, \mathbf{c}_1, o_1\}$ and $\boldsymbol{\theta}_2 = \{\mathbf{q}_2, \mathbf{s}_2, \mathbf{c}_2, o_2\}$ with $\boldsymbol{\theta}_1 \neq \boldsymbol{\theta}_2$, that generate the identical Single Gaussian Radiance Field (SGRF). The identity $L_{\mathcal{G}_1}(\mathbf{x}, \mathbf{d}) = L_{\mathcal{G}_2}(\mathbf{x}, \mathbf{d})$ for all $(\mathbf{x}, \mathbf{d})$ requires two conditions to be met:

1. **Geometric Equivalence**: The covariance matrices must be equal, $\boldsymbol{\Sigma}_1 = \boldsymbol{\Sigma}_2$, which implies that the volume densities $\sigma_{\mathcal{G}}(\mathbf{x})$ are identical. We also assume equal opacity, $o_1 = o_2$.

2. **Appearance Equivalence**: The view-dependent color functions must be identical, which requires $\text{SH}(\mathbf{c}_1, \mathbf{R}_1^\top \mathbf{d}) = \text{SH}(\mathbf{c}_2, \mathbf{R}_2^\top \mathbf{d})$ for all $\mathbf{d} \in \mathbb{S}^2$, where $\mathbf{R}$ is the rotation matrix for the quaternion $\mathbf{q}$.

We construct a distinct parameter set $\boldsymbol{\theta}_2$ by considering a discrete symmetry of the Gaussian ellipsoid. Let $\boldsymbol{\theta}_1 = \{\mathbf{q}_1, \mathbf{s}_1, \mathbf{c}_1, o_1\}$ be an initial parameterization.

Let $\mathbf{R}_{\text{flip}}$ be a rotation matrix corresponding to a 180-degree rotation about one of the local axes (e.g., the z-axis), such that $\mathbf{R}_{\text{flip}} = \text{diag}(-1, -1, 1)$. $\mathbf{R}_{\text{flip}}$ is its own inverse, $\mathbf{R}_{\text{flip}}^\top = \mathbf{R}_{\text{flip}}$. We define a new parameter set $\boldsymbol{\theta}_2$ as follows:

- Let the new rotation be $\mathbf{R}_2 = \mathbf{R}_1 \mathbf{R}_{\text{flip}}$. This defines a new quaternion $\mathbf{q}_2 \neq \mathbf{q}_1$.
- Let the scales and opacity remain unchanged: $\mathbf{s}_2 = \mathbf{s}_1$ and $o_2 = o_1$.

First, we verify geometric equivalence. The new covariance matrix $\boldsymbol{\Sigma}_2$ is:

$$\boldsymbol{\Sigma}_2 = \mathbf{R}_2 \text{diag}(\mathbf{s}_2)^2 \mathbf{R}_2^\top = (\mathbf{R}_1 \mathbf{R}_{\text{flip}}) \text{diag}(\mathbf{s}_1)^2 (\mathbf{R}_1 \mathbf{R}_{\text{flip}})^\top$$
$$= \mathbf{R}_1 \left( \mathbf{R}_{\text{flip}} \text{diag}(\mathbf{s}_1)^2 \mathbf{R}_{\text{flip}}^\top \right) \mathbf{R}_1^\top$$

Since $\mathbf{R}_{\text{flip}}$ is diagonal, it commutes with the diagonal scaling matrix $\text{diag}(\mathbf{s}_1)^2$, meaning the term in parentheses equals $\text{diag}(\mathbf{s}_1)^2$. Therefore,

$$\boldsymbol{\Sigma}_2 = \mathbf{R}_1 \text{diag}(\mathbf{s}_1)^2 \mathbf{R}_1^\top = \boldsymbol{\Sigma}_1.$$

Geometric equivalence is satisfied.

Next, for appearance equivalence, we must find SH coefficients $\mathbf{c}_2$ such that:

$$\text{SH}(\mathbf{c}_1, \mathbf{R}_1^\top \mathbf{d}) = \text{SH}(\mathbf{c}_2, (\mathbf{R}_1 \mathbf{R}_{\text{flip}})^\top \mathbf{d}) = \text{SH}(\mathbf{c}_2, \mathbf{R}_{\text{flip}}^\top \mathbf{R}_1^\top \mathbf{d}).$$

Let $\mathbf{v} = \mathbf{R}_1^\top \mathbf{d}$ be the view direction in the local frame of the first Gaussian. The condition becomes $\text{SH}(\mathbf{c}_1, \mathbf{v}) = \text{SH}(\mathbf{c}_2, \mathbf{R}_{\text{flip}}^\top \mathbf{v})$. This states that the function defined by $\mathbf{c}_2$ when evaluated on a transformed vector must equal the function defined by $\mathbf{c}_1$ on the original vector. This is equivalent to stating that the function itself has been rotated. The properties of spherical harmonics guarantee that

---
[*]Use footnote for providing further information about author (webpage, alternative address)—*not* for acknowledging funding agencies. Funding acknowledgements go at the end of the paper.

for any rotation, there exists a linear transformation (the Wigner D-matrix $\mathbf{D}$) that maps the original coefficients to the new ones. We can therefore find $\mathbf{c}_2$ such that:

$$\mathbf{c}_2 = \mathbf{D}(\mathbf{R}_{\text{flip}})\mathbf{c}_1.$$

We have constructed a parameter set $\boldsymbol{\theta}_2 = \{\mathbf{q}_2, \mathbf{s}_1, \mathbf{c}_2, o_1\}$, which is distinct from $\boldsymbol{\theta}_1$ (since $\mathbf{q}_2 \neq \mathbf{q}_1$) yet defines the identical radiance field. This discrete symmetry exists for any Gaussian, proving the general proposition.

Furthermore, this non-uniqueness expands from a discrete set to a continuous manifold of solutions for symmetric geometries.

- **Case A (Isotropic)**: If $\mathbf{s} = (s, s, s)^\top$, the covariance matrix $\boldsymbol{\Sigma} = s^2\mathbf{I}$ is invariant under *any* rotation. This gives rise to a continuous, multi-parameter family of equivalent solutions.

- **Case B (Spheroidal)**: If two scale components are equal (e.g., $\mathbf{s} = (s_a, s_a, s_b)^\top$), $\boldsymbol{\Sigma}$ is invariant to any rotation around the local axis of symmetry, resulting in a one-parameter continuous family of redundant solutions.

In all such cases, a corresponding transformation on the SH coefficients preserves appearance equivalence. Since non-unique parameterizations exist in all cases, the proposition is proven.

# B  PROOF OF UNIQUENESS OF THE REPRESENTATION $\mathcal{E}_i = (\mathcal{M}_i, F_i)$

Assume $\phi_{\mathcal{G}_1} = \phi_{\mathcal{G}_2}$, namely, $\rho_{\mathcal{G}_1}(\mathbf{x}) = \rho_{\mathcal{G}_2}(\mathbf{x}), \forall \mathbf{x} \in \mathbb{R}^3$ and $c_{\mathcal{G}_1}(\mathbf{d}) = c_{\mathcal{G}_2}(\mathbf{d}), \forall \mathbf{d} \in \mathbb{S}^2$. We show this leads to $\mathcal{E}_1 = \mathcal{E}_2$ for two SGRFs. This implies both $\mathcal{M}_1 = \mathcal{M}_2$ and $F_1 = F_2$.

The volume densities are equal:

$$\exp\left(-\frac{1}{2}(\mathbf{x} - \boldsymbol{\mu}_1)^\top \boldsymbol{\Sigma}_1^{-1}(\mathbf{x} - \boldsymbol{\mu}_1)\right) = \exp\left(-\frac{1}{2}(\mathbf{x} - \boldsymbol{\mu}_2)^\top \boldsymbol{\Sigma}_2^{-1}(\mathbf{x} - \boldsymbol{\mu}_2)\right)$$

Taking the natural logarithm of both sides yields that the quadratic forms are identical for all $\mathbf{x}$:

$$(\mathbf{x} - \boldsymbol{\mu}_1)^\top \boldsymbol{\Sigma}_1^{-1}(\mathbf{x} - \boldsymbol{\mu}_1) = (\mathbf{x} - \boldsymbol{\mu}_2)^\top \boldsymbol{\Sigma}_2^{-1}(\mathbf{x} - \boldsymbol{\mu}_2)$$

An unnormalized Gaussian distribution is uniquely defined by its mean and covariance. Therefore, this equality implies $\boldsymbol{\mu}_1 = \boldsymbol{\mu}_2$ and $\boldsymbol{\Sigma}_1 = \boldsymbol{\Sigma}_2$.

The submanifold $\mathcal{M}_i$ is defined as the level set:

$$\mathcal{M}_i = \{\mathbf{x} \in \mathbb{R}^3 \mid (\mathbf{x} - \boldsymbol{\mu}_i)^\top \boldsymbol{\Sigma}_i^{-1}(\mathbf{x} - \boldsymbol{\mu}_i) = r^2\}$$

Since the parameters $(\boldsymbol{\mu}_i, \boldsymbol{\Sigma}_i)$ that define the level set are identical for both primitives, the resulting sets of points must also be identical. Thus, $\mathcal{M}_1 = \mathcal{M}_2$.

The submanifold color field $F_i$ is defined for a point $\mathbf{x} \in \mathcal{M}_i$ as:

$$F_i(\mathbf{x}) = c_{\mathcal{G}_i}(\mathbf{d_x}), \quad \text{where} \quad \mathbf{d_x} = (\mathbf{x} - \boldsymbol{\mu}_i)/\|\mathbf{x} - \boldsymbol{\mu}_i\|$$

From the hypothesis, we know that $c_{\mathcal{G}_1}(\mathbf{d}) = c_{\mathcal{G}_2}(\mathbf{d})$ holds for any unit direction vector $\mathbf{d} \in \mathbb{S}^2$.

For any point $\mathbf{x}$ on the common manifold $\mathcal{M} = \mathcal{M}_1 = \mathcal{M}_2$, its corresponding direction vector $\mathbf{d_x}$ is an element of $\mathbb{S}^2$. We can therefore apply the hypothesis for this specific direction:

$$c_{\mathcal{G}_1}(\mathbf{d_x}) = c_{\mathcal{G}_2}(\mathbf{d_x})$$

By the definition of $F_i$, this directly implies:

$$F_1(\mathbf{x}) = F_2(\mathbf{x})$$

This holds for all $\mathbf{x} \in \mathcal{M}$. Thus, the color fields $F_1$ and $F_2$ are identical.

## C   IMPLEMENTATION DETAILS

### C.1   SUBMANIFOLD FIELD VAE

For completeness, we note a few aspects not detailed in the main text. Also see Alg. **??** for steps in one training step.

**Uniform Point Sampling.** The submanifold field $\mathcal{E}_i$ is discretized by using a uniform mesh grid of size $(n, n)$ to sample $P = n^2$ points on the ellipsoidal surface $\mathcal{M}_i$ with respect to area, forming $\mathcal{P}_i = \{(\mathbf{x}_{i,k}, F_i(\mathbf{x}_{i,k}), \alpha_i)\}_{k=1}^P$.

**Decoding Gaussian Parameters.** After decoding, we recover Gaussian parameters from the reconstructed point cloud by first applying batched PCA to estimate the ellipsoid axes and orientation: we compute the mean and covariance of the points, perform eigen decomposition to obtain principal axes, and ensure a right-handed coordinate system. The logarithm of the axis lengths gives the scale parameters, and the rotation matrix is converted to a quaternion using a numerically stable batched algorithm. For appearance, we compute ellipsoid-normalized directions for each point and fit spherical harmonics coefficients to the RGB values via regularized batched least-squares. Opacity is estimated by averaging and logit-transforming the per-point values.

---

**Algorithm 1** SF-VAE: one training step for a minibatch of submanifold fields

---

**Require:** Batch $\{(\boldsymbol{\mu}_i, \boldsymbol{\Sigma}_i, \mathbf{c}_i, o_i)\}_{i=1}^B$, fixed $r^2 = 1$, point count $P$
 1: **for** each $i$ **do**
 2:     *(Sampling on $\mathcal{M}_i$)* Sample $\{\mathbf{u}_k\}_{k=1}^P$ quasi-uniformly on $\mathbb{S}^2$; set $\mathbf{x}_{i,k} \leftarrow \boldsymbol{\mu}_i + \boldsymbol{\Sigma}_i^{1/2}\mathbf{u}_k$ so that $(\mathbf{x}_{i,k} - \boldsymbol{\mu}_i)^\top \boldsymbol{\Sigma}_i^{-1}(\mathbf{x}_{i,k} - \boldsymbol{\mu}_i) = r^2$
 3:     *(Color/opacity)* $\mathbf{d}_{i,k} \leftarrow (\mathbf{x}_{i,k} - \boldsymbol{\mu}_i)/\|\mathbf{x}_{i,k} - \boldsymbol{\mu}_i\|$;   $\mathbf{c}_{i,k} \leftarrow \mathrm{SH}(\mathbf{c}_i, \mathbf{d}_{i,k})$;   $\alpha_{i,k} \leftarrow \sigma(o_i)$
 4:     *(Point set)* $\mathcal{P}_i \leftarrow \{(\mathbf{x}_{i,k}, \mathbf{c}_{i,k}, \alpha_{i,k})\}_{k=1}^P$
 5: **end for**
 6: *(Encode)* $(\boldsymbol{\mu}_i^z, \log \boldsymbol{\sigma}_i^{2,z}) \leftarrow E(\mathcal{P}_i)$;  $\mathbf{z}_i \leftarrow \boldsymbol{\mu}_i^z + \boldsymbol{\sigma}_i^z \odot \varepsilon$, $\varepsilon \sim \mathcal{N}(0, I)$
 7: *(Decode)* $(\widehat{\mathcal{M}}_i, \widehat{F}_i, \hat{\alpha}_i) \leftarrow D(\mathbf{z}_i)$;  $\widehat{\mathcal{P}}_i \leftarrow \{(\hat{\mathbf{x}}_{i,k}, \widehat{F}_i(\hat{\mathbf{x}}_{i,k}), \hat{\alpha}_i)\}$
 8: *(Recover parameters)* $\widehat{\boldsymbol{\Sigma}}_i \leftarrow \mathrm{PCA}(\{\hat{\mathbf{x}}_{i,k}\})$;  $\widehat{\mathbf{c}}_i \leftarrow \arg\min_{\mathbf{c}} \sum_k \|\mathrm{SH}(\mathbf{c}, \hat{\mathbf{d}}_{i,k}) - \widehat{F}_i(\hat{\mathbf{x}}_{i,k})\|_2^2$
 9: *(Loss)* $\mathcal{L}_{\mathrm{rec}} \leftarrow W_2^{(\varepsilon)}(\mathcal{P}_i, \widehat{\mathcal{P}}_i)$ with $d(\cdot, \cdot)$ from Eq. (8);  $\mathcal{L} \leftarrow \mathcal{L}_{\mathrm{rec}} + \beta \cdot \mathrm{KL}$
10: *(Update)* $\boldsymbol{\theta}_E, \boldsymbol{\theta}_D \leftarrow \boldsymbol{\theta}_E, \boldsymbol{\theta}_D - \eta \nabla_{\boldsymbol{\theta}_E, \boldsymbol{\theta}_D} \mathcal{L}$

---

### C.2   GENERATED 3D GAUSSIAN PRIMITIVES DATASET

**Parameter priors and sampling** Each primitive $\mathcal{G}_i$ is sampled as $\boldsymbol{\theta}_i = \{\boldsymbol{\mu}_i, \mathbf{q}_i, \mathbf{s}_i, \mathbf{c}_i, o_i\}$ and converted to $(\boldsymbol{\mu}_i, \boldsymbol{\Sigma}_i, \mathbf{c}_i, \alpha_i)$ with $\boldsymbol{\Sigma}_i = R(\mathbf{q}_i) \mathrm{diag}(\exp(\mathbf{s}_i))^2 R(\mathbf{q}_i)^\top$ and $\alpha_i = \sigma(o_i)$. Unless otherwise stated we use:

- **Mean.** $\boldsymbol{\mu}_i = (0, 0, 0)$. Since our setting only samples single Gaussians, extrinsic information is not needed.

- **Rotation.** $\mathbf{q}_i$ is sampled uniformly on $SO(3)$ (normalized and, if needed, enforce a canonical sign).

- **Scale.** Log–axes $\mathbf{s}_i \in \mathbb{R}^3$ drawn i.i.d. from $\mathcal{U}([s_{min}, s_{max}])$; set activated scales $\exp(s_i)$.

- **SH coefficients.** Let $\beta > 1$ be the decay factor (in code, $\beta = 4$). For degree $\ell = 0, \ldots, L$, we draw the $(2\ell+1)$-dimensional SH band as $\mathbf{c}_\ell \sim \mathcal{N}(\mathbf{0}, \sigma_\ell^2 I_{2\ell+1})$,    $\sigma_\ell = \beta^{-\ell}$, i.e., $\mathrm{Var}[c_{\ell,m}] = \beta^{-2\ell}$ for $m = -\ell, \ldots, \ell$. If coefficients above the chosen degree $L$ are padded up to $L_{\max}$, we use i.i.d. noise $c_{\ell,m} \sim \mathcal{N}(0, \sigma_{\mathrm{void}}^2)$ with $\sigma_{\mathrm{void}} = 0.05$.

- **Opacity.** Logit $o_i \sim \mathcal{U}([o_{min}, o_{max}])$, with $\alpha_i = \sigma(o_i)$.

---

**Algorithm 2** GAUSSIANGEN: Generate one colored point cloud from a random Gaussian

---

**Require:** point count $P$ (default 144), SH degree $L_{\max}$ with $K = (L_{\max}+1)^2$
1: **Sample raw parameters** $\boldsymbol{\mu} \in \mathbb{R}^3$, $\mathbf{s} \in \mathbb{R}^3$, $\mathbf{q} \in SO(3)$, $o \in \mathbb{R}$, feat_dc $\in \mathbb{R}^3$, feat_extra $\in \mathbb{R}^{3(K-1)}$
2: **Assemble SH coefficients** $\mathbf{C} \in \mathbb{R}^{3 \times K}$ by stacking per channel:
   $\mathbf{C}_r \leftarrow [\text{feat\_dc}[0], \text{ feat\_extra}[0 : (K-1)]]$
   $\mathbf{C}_g \leftarrow [\text{feat\_dc}[1], \text{ feat\_extra}[(K-1) : 2(K-1)]]$
   $\mathbf{C}_b \leftarrow [\text{feat\_dc}[2], \text{ feat\_extra}[2(K-1) : 3(K-1)]]$
3: **Activate parameters**
   $\mathbf{q} \leftarrow \mathbf{q}/\|\mathbf{q}\|$; $R \leftarrow R(\mathbf{q})$; $\boldsymbol{\sigma} \leftarrow \exp(\mathbf{s})$; $\alpha \leftarrow \sigma(o)$
4: **Build surface grid**
   $n \leftarrow \sqrt{P}$; create angular grids $u \in [0, 2\pi)$ and $v \in [0, \pi]$ of size $n \times n$
   directions $\mathbf{d}(u,v) \in \mathbb{S}^2$ from spherical angles $(u, v)$
5: **Map to ellipsoid (iso-density $r^2{=}1$)**
   $\mathbf{x}(u,v) \leftarrow \boldsymbol{\mu} + R \, \text{diag}(\boldsymbol{\sigma}) \, \mathbf{d}(u,v)$
6: **Evaluate color field**
   $\mathbf{c}(u,v) \leftarrow \text{SH}(\mathbf{C}, \mathbf{d}(u,v))$
   *(optional)* color post-process: $\mathbf{c} \leftarrow \text{clip}(\mathbf{c}+0.5, \, 0, 1)$
7: **Pack output**
   replicate $\alpha$ to all points; flatten grids to $P$ points
   **return** $\{(\mathbf{x}_k, \, \mathbf{c}_k, \, \alpha)\}_{k=1}^P \in \mathbb{R}^{P \times 7}$ // $(x, y, z, r, g, b, \alpha)$

---

To sum up, we can describe the data distribution in this dataset as:

$$\mathcal{D} = \left\{ (\boldsymbol{\mu}_i, \mathbf{q}_i, \mathbf{s}_i, \mathbf{c}_i, o_i) \right\}_{i=1}^N \stackrel{\text{i.i.d.}}{\sim} \underbrace{\delta_{\mathbf{0}}}_{\boldsymbol{\mu}} \times \underbrace{\mathcal{U}(SO(3))}_{\mathbf{q}} \times \underbrace{\mathcal{U}([s_{\min}, s_{\max}])^3}_{\mathbf{s}}$$

$$\times \underbrace{\left( \prod_{c \in \{R,G,B\}} \Big[ \prod_{\ell=0}^{L} \mathcal{N}(0, \beta^{-2\ell})^{2\ell+1} \times \prod_{\ell=L+1}^{L_{\max}} \mathcal{N}(0, \sigma_{\text{void}}^2)^{2\ell+1} \Big] \right)}_{\mathbf{c} \in \mathbb{R}^{3 \times K}, \; K=(L_{\max}+1)^2}$$

$$\times \underbrace{\mathcal{U}([o_{\min}, o_{\max}])}_{o}, \qquad \beta > 1 \text{ (default 4)}, \; \sigma_{\text{void}} = 0.05.$$

where $\beta > 1$ is the SH variance–decay factor (default $\beta{=}4$) and $\sigma_{\text{void}}$ is the padding noise std (default 0.05). Activations used downstream are $\Sigma_i = R(\mathbf{q}_i) \, \text{diag}(\exp(\mathbf{s}_i))^2 R(\mathbf{q}_i)^\top$ and $\alpha_i = \sigma(o_i)$.

**Defaults and ablations** Default hyperparameters: $N{=}500K$, $L_{\max}{=}3$ ($K{=}16$), $P{=}144$, $s_{min}{=}-8$, $s_{max}{=}0$, $o_{min}{=}-5$, $o_{max}{=}10$. These parameters are built from statistical analysis of data distribution in diverse 3DGS datasets. We ablate $P \in \{36, 64, 144, 576\}$ in the main paper to assess reconstruction vs. cost.

**Data Formatting and Processing.** For each of the randomly synthesized primitive we store the native tuple $\boldsymbol{\theta}_i$ (float32 arrays for $\boldsymbol{\mu}$, $\mathbf{q}$, $\mathbf{s}$, $\mathbf{c}$, $o$). The discretized field $\mathcal{P}_i$ is sampled from the primitive of $\boldsymbol{\theta}_i$ to a tensor of $(B, P, 7)$ where 7 is $(x, y, z, r, g, b, \alpha)$. It can be obtained at runtime to reduce memory and storage requirements. The dataset exposes toggles to return either $\boldsymbol{\theta}$ or $\mathcal{P}$.

# D ADDITIONAL EVALUATION AND VISUALIZATION RESULTS

## D.1 INTERPOLATION & NOISE VISUALIZATIONS

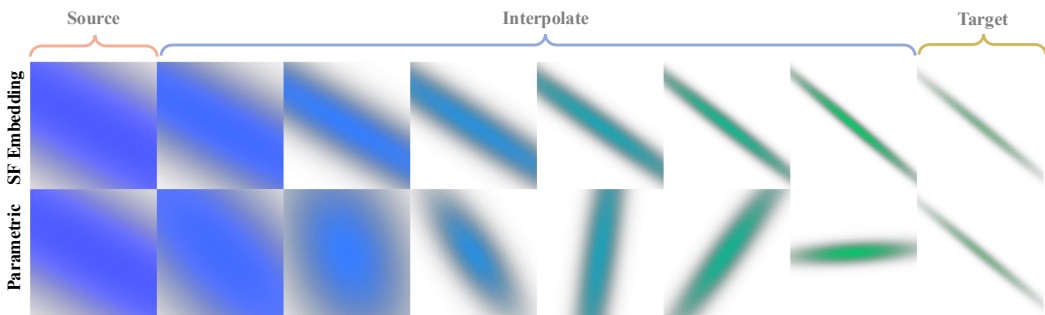

Figure a: Sample visual comparison of linear interpolation in submanifold field embedding space and parametric space. Interpolation in SF embedding shows smooth transition from source to target. Perturb **s** and **q** will dramatically change the geometry of the resulted Gaussian, verifying the feature heterogeneous problem.

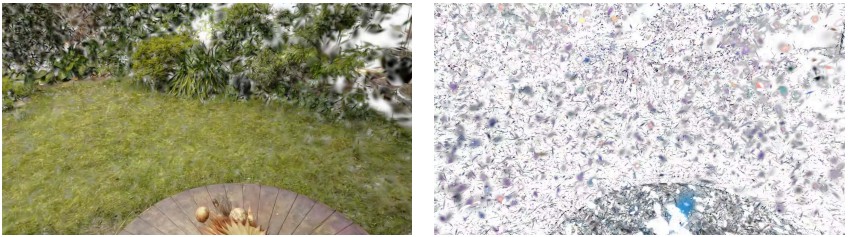

Figure b: When rotation **q** is inverted to −**q**, the submanifold field model (left) can still correctly reconstruct the scene, while parametric model fails to process this equivalent rotation.

## D.2 Additional Reconstruction Visualizations

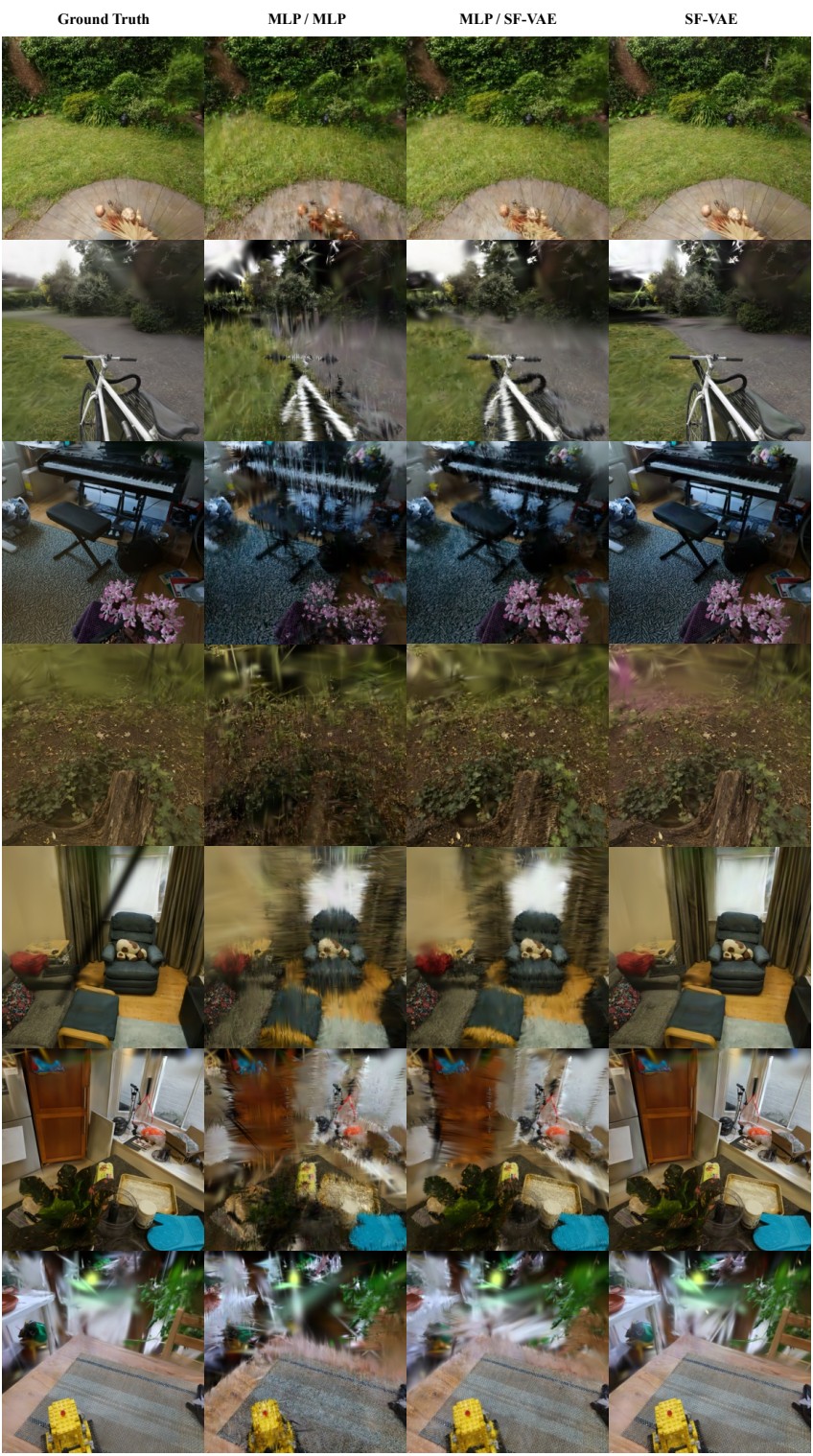

Figure c: Full qualitative results for rasterization using different model configurations on Mip-NeRF 360.

### D.3 UNSUPERVISED CLUSTERING QUANTITATIVE RESULTS

| Clustered Feature | Silhouette ↑ | Cluster Compactness (MSE ↓) |
|---|---|---|
| Parameters | 0.090 | 71.306 |
| MLP / MLP | 0.084 | 50.761 |
| MLP / SF-VAE | 0.086 | 50.916 |
| SF-VAE | **0.113** | **48.282** |

Table a: Quantitative comparisons on unsupervised clustering based on the respective clustered feature. Higher silhouette is better; lower compactness (MSE) is better.

### D.4 GAUSSIAN NEURAL FIELD DETAILS & VISUALIZATIONS

**Network Structure for Gaussian Neural Fields.** We define a compact per-scene MLP that takes only 3D coordinates ((x,y,z)), applies sinusoidal/Fourier positional encoding, and passes the encoded vector through a fully connected backbone (256-width 4 layers for ShapeSplat, 512-width 8 layers for Mip-NeRF 360) with a single mid-network skip that re-concatenates the encoded input for better conditioning. The *same network architecture* is used for both targets; the only difference is the final output dimension: either 32 for the SF embedding or 56 for the no-position Gaussian parameters ($[q(4), s(3), \text{SH}(3K), o(1)]$). Hidden layers use standard ReLU/SiLU activations, the output is linear, and the overall footprint remains lightweight ($\approx 2 \times 10^5$ parameters for ShapeSplat, $\approx 1.8 \times 10^6$ for Mip-NeRF 360) for querying at Gaussian centers to yield SF embeddings or directly renderable Gaussian parameters.

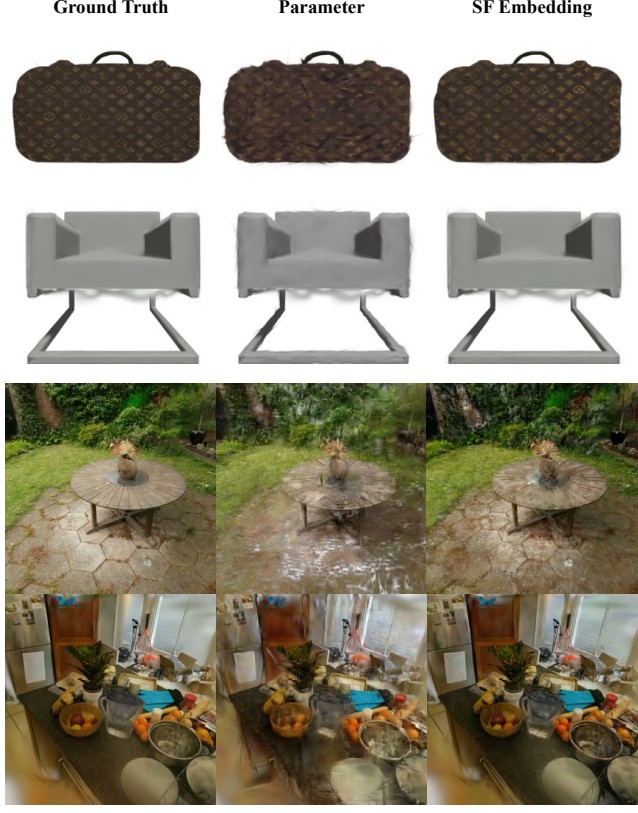

Figure d: Arbitrarily picked visual comparisons of Gaussian Neural Fields trained with target Gaussian parameters and submanifold field embeddings. The submanifold field embedding results is qualitatively better under matching settings, indicating that the SF embedding is a better learning target for regression tasks such as training a neural field.