# OpenReview forum: "Learning Unified Representation of 3D Gaussian Splatting"
_ICLR.cc/2026/Conference — ICLR 2026 Poster_

### Official Review · Reviewer_jRea · 2025-10-26

**Soundness:** 2
**Presentation:** 3
**Contribution:** 2
**Rating:** 2
**Confidence:** 4

**Summary:**

This paper addresses challenges in 3D Gaussian Splatting (3DGS) representations, where individual Gaussians have multiple parameter sets. The authors note that some parameters are non-unique (e.g., different rotation representations can yield the same state) and that parameter groups are from different distributions or manifolds. The authors claim this non-uniqueness and heterogeneity make the standard 3DGS parameter space ill-suited for neural networks tasked with generating or estimating Gaussians. To resolve this, the paper introduces "submanifold field" representations, a novel parameter space for these Gaussian features. The authors demonstrate the uniqueness of this new representation and propose a corresponding manifold distance (M-Dist).
Through experiments, the paper demonstrated superior performance against baselines.

**Strengths:**

- The paper presents an interesting and timely idea, given the growing interest in large-scale reconstruction models which could benefit from improved 3D Gaussian Splatting representations.
- The proposed method demonstrates strong performance improvements over the baseline.
- The authors provide a broad analysis, including evaluations of robustness, clustering, and interpolation.

**Weaknesses:**

**Motivation**: The motivation presented in the Introduction could be strengthened. The importance of resolving the identified issues should be emphasized more clearly, particularly by highlighting specific downstream applications and detailing why and when existing representations fail.

**Experiments**: The current experiments focus on small MLP and VAE-style networks, which may not reflect real-world applications. Even if the goal is to isolate the effect of the new representation, it is unclear whether results from these small networks will generalize to the much larger, more complex models used in practice. Without testing or adapting existing large-scale reconstruction models, the broader applicability of the proposed representation remains uncertain.

**Questions:**

- To aid in understanding the concept, could the authors elaborate on the definitions and intentions behind the terms "sub" and "field" in "submanifold fields"?

- While the paper discusses the ill-posedness of standard 3DGS features, is there a specific reason to formalize the non-uniqueness and heterogeneity so strongly through definitions and propositions?

- In Section 4.2 (lines 362-364), it is stated that M-Dist has a higher correlation with the perceptual metric than L1. Could the authors also provide the correlation with L2? Furthermore, in Figure 4 (right table), the M-Dist for the SF embedding appears to increase significantly with noise. This seems to contradict the claim that M-Dist has a better correlation with perceived image quality than L1. Could the authors clarify this apparent discrepancy?

- Regarding Figure 4 (robustness to noise): Since the different representations have different scales and properties, is it certain that adding uniform noise affects them in a comparable way?

- For Figure 5 (interpolation): Would it be possible to include more straightforward or intuitive examples? From the current figure, it is unclear precisely how the SF-VAE interpolation is superior to the other methods.

---

> ### Author Response · Authors · 2025-11-26
> **To Reviewer jRea (1/2)**
>
> We thank **Reviewer jRea** for the evaluation of our work and for raising several points regarding motivation, applicability, and clarity. First, we would like to rephrase this problem and our contribution as follows:
>
> 1. **Problem: non-uniqueness and heterogeneity for a single Gaussian splat.**
>
>     Neural networks that operate directly in the raw Gaussian parameter space suffer fundamental failures, leading to **embedding collisions**. A clear example is Appendix Fig. 9: the network fails completely even though ($q$) and ($-q$) represent the same Gaussian. Another example appears under small parameter perturbations: **minor changes** can cause **large distortions**. Such small perturbations can arise from many sources, including direct noise and domain shift during learning.
>
> 2. **Solution: a new framework that provably avoids these failures and yields a smoother embedding space.**
>
>     Our formulation enforces uniqueness and regularity by design, producing a **smooth, well-behaved** representation. Although these are solid theoretical contributions, we also emphasize that a **smooth and uniform feature space** has many practical benefits: it serves as a strong target for neural decoders, provides good clustering bases **without extra training**, and is likely to benefit many downstream applications. In our applicability studies, we validate the semantic benefits of our approach with an experiment on **unsupervised graph clustering**, hinting at potential in general retrieval and classification tasks; and through comparative experiments with **neural fields**, we show its great potential in neural decoding and neural compression, and more generally regression-based learning on 3DGS.
>
> In this paper, however, we stop at **defining the problem**, **presenting the solution**, and **providing strong evidence of its potential**. A broader exploration of downstream tasks belongs to a more application-oriented discussion and is outside our current scope.
>
> ---
>
> **W1**: *“… the motivation presented in the Introduction could be strengthened … particularly by highlighting specific downstream applications …”*
>
> A: We have rephrased the introduction section in our paper to reflect this, please see L033 to L089 in the latest version of our paper.
>
> Moreover, we would like to stress again that the theoretical motivations of parametric non-uniqueness and heterogeneity issues are very fundamental in neural networks as every successful neural architecture, from the simplest MLPs to the largest Transformer-based models, heavily depend on the underlying assumption that its dataset provides a stable one-to-one mapping. And as we show in the paper, the current parameter-based data clearly violates this assumption. In our paper, we identify this fundamental problem and provide a solution for the 3DGS community. To our knowledge, no prior work in 3DGS has formally identified this problem, which we consider to be fundamental in modern deep learning. In addition, we provide practical examples to explain why these theoretical concerns might affect downstream applications. Consider one extreme case detailed in Appendix Fig. 9, where flipping the signs for the rotation quaternions resulted in complete failure of embedding and reconstruction. This is further indicated in the new experiment we added, where we trained neural fields for raw parameter data and submanifold field embedding data. In this experiment, we have observed that the SF embedding data is much easier to train on, where it converged faster and have significantly better training outcomes (as we show in the results tables below, also see L470 to L480 in the paper). This demonstrates that submanifold field embedding is a good learning target for neural generation/reconstruction, thus hinting at its potential for generative modeling and neural compression.
>
> | Target |  PSNR ↑  | SSIM ↑  | LPIPS ↓  | # Params |
> | --- | --- | --- | --- | --- |
> | Raw GS Parameter  | 51.660 | 0.925 | 0.141 | 0.21M |
> | SF Embedding  | 58.619 | 0.980 | 0.043 | 0.20M |
>
> | Target |  PSNR ↑  | SSIM ↑  | LPIPS ↓  | # Params |
> | --- | --- | --- | --- | --- |
> | Raw GS Parameter  | 19.922  | 0.648  | 0.410 | 1.87M |
> | SF Embedding  | 24.395 | 0.804 | 0.261 | 1.85M |
>
> Comparison between Gaussian Neural Fields trained using submanifold field embeddings and raw Gaussian parameters. **Top: ShapeSplat, bottom: Mip-NeRF 360**, see Appendix Fig. 11 for visualization.

---

> ### Author Response · Authors · 2025-11-26
> **To Reviewer jRea (2/2)**
>
> **W2**: *“… the current experiments focus on small MLP and VAE-style networks … it is unclear whether results will generalize to larger models …”*
>
> A: To clarify our setting, we independently embed each Gaussian primitive. The size of the network will not affect much towards the encoding of a single Gaussian, as we have demonstrated in the ablation study where we replace our PointNet encoder with DGCNN, a larger encoder with more parameters, yielding no significant difference in performance (see L484 to L496). While larger networks could downgrade the usefulness of the embeddings due to excess computational costs.
>
> ---
>
> **Q1**: *“… could the authors elaborate on the definitions and intentions behind the terms ‘sub’ and ‘field’ …”*
>
> A: “Submanifolds” refers to a lower-dimensional manifold embedded in a higher-dimensional space （i.e., the iso-probability surface in Equation (4)), while “field” means a quantity function defined on it (i.e., the field function in Equation (5) ); thus, “submanifold fields” are fields defined on submanifolds. They are well-established concepts in geometry learning as showcased in [1] [2] .
>
> ---
>
> **Q2**: *“… is there a specific reason to formalize the non-uniqueness and heterogeneity so strongly through definitions and propositions …”*
>
> A: We formalize the non-uniqueness and heterogeneity to highlight the current limitations of the GS parameterization framework and the need for a better representation.
>
> 1. The proposal and rigorous formalization of a new research problem is itself a theoretical contribution.
> 2. Our definition of *non-uniqueness* is essential, as it directly leads to Proposition 2 and the resulting embedding method with theoretical guarantees. These theoretical works lay out a solid foundation for justifying the proposed work, so we believe it is needed and strengthen the rigor of our proposed work.
>
> ---
>
> **Q3**: *“… could the authors provide the correlation with L2 … and clarify the discrepancy where M-Dist increases significantly with noise …”*
>
> A: We want to highlight that Lp (for p>1) distance works similarly to L1 distance in high-dimensional space and will bring similar comparison results with just different calculation values [3].
>
> Furthermore, we indicate that L1 distance is the one that is typically used for optimizing Gaussian Splatting parameters, such as the original 3DGS paper [4], generative models like DiffSplat [5] and many more works. Thus, it makes more sense for us to include L1 distance in the comparison instead of L2.
>
> ---
>
> **Q4**: *“… M-Dist for the SF embedding appears to increase significantly with noise. This seems to contradict the claim that M-Dist has a better correlation with perceived image quality than L1 …”*
>
> A: We apologize for the confusion here. We make a mistake when labeling the figure. This has now been corrected in the latest version of our paper.
>
> ---
>
> **Q5**: *“… would it be possible to include more straightforward or intuitive interpolation examples in Figure 5 …”*
>
> A: We are assuming that the reviewer is referring to Fig. a in the original appendix, we would like to explain this figure a bit more. The parameter space interpolation clearly shows some unnecessary rotation and scale changes compared with the SF embedding interpolation. As you can see in this sample visualization we provide, from source to target, the general trend is that rotation stays relatively unchanged, while scale and color shifts smoothly. This is not the case with parameter interpolation. The unnecessary jitter in rotation and scale suggests that the parametric space is not numerically smooth and thus sensitive to random perturbation. This type of issue is not ideal for downstream regression tasks as neural networks’ feedforwarding always induce numerical noises.
>
> ---
>
> [1] Graham, Benjamin, Martin Engelcke, and Laurens Van Der Maaten. "3d semantic segmentation with submanifold sparse convolutional networks." Proceedings of the IEEE conference on computer vision and pattern recognition. 2018.
>
> [2] Xie, Yiheng, et al. "Neural fields in visual computing and beyond." Computer graphics forum. Vol. 41. No. 2. 2022.
>
> [3] Angiulli, Fabrizio. "On the behavior of intrinsically high-dimensional spaces: distances, direct and reverse nearest neighbors, and hubness." *Journal of Machine Learning Research* 18.170 (2018): 1-60.
>
> [4] Bernhard Kerbl, Georgios Kopanas, Thomas Leimk¨uhler, and George Drettakis. 3d gaussian splatting for real-time radiance field rendering. ACM Trans. Graph., 2023.
>
> [5] Lin C, Pan P, Yang B, et al. DiffSplat: Repurposing Image Diffusion Models for Scalable Gaussian Splat Generation. International Conference on Learning Representations (ICLR), 2025.

---

> > ### Comment · Reviewer_jRea · 2025-11-27
> >
> > I appreciate the authors’ efforts in the rebuttal and their clarifications regarding the paper’s goals. The problem formulation around non-uniqueness and heterogeneity in existing 3D Gaussian Splatting representations is meaningful, and the proposed geometric perspective is novel and interesting. However, I would still like to maintain my original assessment for the following reason.
> >
> > **Conceptual scope and framing.**
> > Throughout the paper, two different threads appear to be mixed.
> > One thread concerns defining a new 3D Gaussian Splatting representation with desirable properties such as uniqueness and channel homogeneity.
> > The other concerns how to feed Gaussian parameters into neural networks, specifically auto-encoder–style models that require an encoder.
> > The paper appears to treat these two issues as equivalent, but from my perspective they are not. This is also why both reviewer Jmon and I expressed concerns about applicability: **the proposed formulation does not act as a drop-in alternative representation (e.g., analogous to comparing sRGB vs. HSV), but instead introduces a processing pipeline for encoding Gaussian parameters.** The method relies on sampling each Gaussian into a point-cloud-like structure and feeding it to architectures that are friendly to point set encoders. Consequently, the experimental evaluation focuses primarily on how to process Gaussian parameters for an encoder, rather than evaluating a new representation that can directly replace existing 3DGS representations.
> >
> > If the primary contribution were indeed a new representation intended to supersede existing Gaussian features, the necessity of the auto-encoder framework would be unclear. Conversely, if the primary contribution were a new encoding pipeline for Gaussian-based inputs, then the experimental setup should target tasks where Gaussian features are actually taken as input and processed by a network, rather than relying solely on reconstruction-based VAE experiments. While I understand the authors’ position that they aim to “provide strong evidence of potential,” I still find a discrepancy between the conceptual motivation and the method and experiments. The proposed submanifold field exists only as a sampled structure during encoding and does not function as a stand-alone representation in downstream 3DGS pipelines, reinforcing that the method is closely tied to encoder-specific processing rather than a unified alternative representation.
> >
> > I continue to agree that more principled representations for 3D Gaussian Splatting would be valuable, and the paper contains interesting ideas in this direction. However, in its current form, **the motivation, formulation, and experimental design do not fully align**, and therefore I maintain my original score.

---

> > > ### Author Response · Authors · 2025-12-01
> > > **Reply to Reviewer jRea**
> > >
> > > We appreciate the reviewer’s comments. Until the latest comments, we just understood there might be a misunderstanding regarding the scope and main contribution of our work. The proposed SF embedding, once trained, is indeed serving as a *drop-in replacement* for the original 3DGS parameters.
> > >
> > > Here we do not claim to replace the raw GS parameters completely, as they are still used for rendering. We claim that we *train and fix an encoder, and then use its output embedding to substitute raw GS parameters* for downstream tasks *except for rendering* (e.g., segmentation, generation, compression). And our primary contribution is to theoretically and experimentally demonstrate that this new representation is better than parametric one with small information loss.
> > >
> > > Highlights:
> > >
> > > Theoretically, we demonstrate:
> > >
> > > - The **formulation of issues** with the parametric representation in Section 3.2 and Proposition 1
> > > - The **uniqueness of the submanifold field representation** in Proposition 2, Section 3.3
> > >
> > > Experimentally, we demonstrate:
> > >
> > > - **Generalizability**: SF embedding is good at preserving visual information with best generalizations across datasets, see Table 1 and Table 2
> > > - **Smoothness**: SF embedding space is smooth and robust, see Figure 4 and Appendix Figure 8
> > > - **Downstream applicability**:
> > >     1. Clustering/segmentation. Figure 5 **(add results)** and Appendix Table 4 **(new results)**
> > >     2. Neural Field Compression/Decoder. Table 3 **(new results)** and Appendix Figure 11 **(new results)**
> > >
> > > We display some results referenced above:
> > >
> > > Table 3: Comparison between Gaussian Neural Fields trained using submanifold field embeddings and raw Gaussian parameters.
> > >
> > > | Dataset | Target | PSNR ↑ | SSIM ↑ | LPIPS ↓ | # Params |
> > > | --- | --- | --- | --- | --- | --- |
> > > | ShapeSplat | Raw GS Parameter | 51.660 | 0.925 | 0.141 | 0.21M |
> > > | ShapeSplat | SF Embedding | **58.619** | **0.980** | **0.043** | 0.20M |
> > > | Mip-NeRF 360 | Raw GS Parameter | 19.922 | 0.648 | 0.410 | 1.87M |
> > > | Mip-NeRF 360 | SF Embedding | **24.395** | **0.804** | **0.261** | 1.85M |
> > >
> > > Table 4: Quantitative comparisons on unsupervised clustering based on the respective clustered
> > > feature.
> > >
> > > | Clustered Feature | Silhouette ↑ | Cluster Compactness (MSE) ↓ |
> > > | --- | --- | --- |
> > > | Parameters | 0.090 | 71.306 |
> > > | MLP / MLP | 0.084 | 50.761 |
> > > | MLP / SF-VAE | 0.086 | 50.916 |
> > > | SF-VAE | **0.113** | **48.282** |

---

### Official Review · Reviewer_Jmon · 2025-10-31

**Soundness:** 3
**Presentation:** 3
**Contribution:** 3
**Rating:** 4
**Confidence:** 3

**Summary:**

This paper addresses key challenges of using parameter-based Gaussian representations as network inputs: non-uniqueness, numerical heterogeneity, and distinct mathematical manifolds. To overcome these issues while preserving the underlying geometric structure, the authors propose representing each Gaussian as a continuous field defined on its iso-probability surface. Experimental results demonstrate that the proposed representation achieves superior reconstruction quality and exhibits strong robustness to domain shifts compared to traditional parametric representations.

**Strengths:**

- The paper clearly explains the problem formulation and the motivations behind introducing a new representation for Gaussian primitives.
- The idea of learning a discretized iso-probability surface as a geometric representation is interesting, and I believe it is also novel.
- The writing is clear and effectively conveys the core ideas; the main figure is particularly helpful in illustrating the concept.
- The proposed model can be trained using a randomly generated dataset, enabling domain-agnostic learning that can later be applied to embed optimized Gaussian primitives.

**Weaknesses:**

- While the paper aims to validate the idea of using iso-probability surfaces as an alternative representation of parameter-based Gaussian representations to make them more amenable to learning-based systems, it remains unclear how well this representation generalizes to various downstream tasks. From my understanding, one motivation for learning such representations is to enable applications such as compressing large sets of Gaussians or training generative models to sample new Gaussian primitives. However, the experimental results primarily demonstrate that the proposed representation can reconstruct Gaussian parameters more accurately in a controlled setting. This alone does not necessarily prove that the learned representation will be effective in downstream or scene-level applications.
- Each iso-probability surface is independently encoded and decoded by the network, meaning that each latent embedding is learned in isolation without awareness of global scene-level context. This independence could limit the model’s ability to generate spatially consistent or semantically coherent Gaussian primitives when generating full scenes, which is an important consideration for practical use cases (e.g., scene generation).

**Questions:**

- The decoding process involves batched PCA, eigen decomposition, and spherical harmonics fitting for every reconstructed Gaussian primitive. Could the authors clarify the computational cost of this step and whether it poses a bottleneck when decoding a large number of Gaussians?

---

> ### Author Response · Authors · 2025-11-26
> **To Reviewer Jmon (1/2)**
>
> We thank **Reviewer Jmon** for the clear evaluation of our method and for pointing out important aspects regarding potentials in downstream tasks and computational cost. We address each point in detail as follows.
>
> ---
>
> **W1**: *“… it remains unclear how well this representation generalizes to various downstream tasks …”*
>
> A:  We firstly indicate that due to time/space constraints, we have focused on presenting the core ideas of the SF embedding, thus we stop at **defining the problem**, **presenting the solution**, and **providing strong evidence of its potential**. A broader exploration of diverse downstream tasks belongs to a more application-oriented discussion and is outside our current scope.
>
> However, we still attempt to present the potential applicability of our embedding in a general manner (we consider them as “midstream” tasks). Two experiments are demonstrated in the updated manuscript (L461 to L480).
>
> 1. Experiment on **unsupervised graph clustering** showcases the **semantic richness**  of the submanifold field embedding space compared to the parametric representation **(also raised by Reviewer kCGA )**. It can produce clean and semantically meaningful segmentation results without supervision, indicating potential gain in learning-based semantic segmentation, and more generally, tasks involving **object retrieval and scene editing**.
>
> 2. We also add a new experiment on **Gaussian Neural Fields.** Particularly, we fit a GS scene/object by a MLP-based neural field of SF-embedding/parameters and our embeddings can work better than the raw GS parameters. This demonstrates that submanifold field embedding is a good learning target for neural generation/reconstruction, thus hinting at its potential for **generative modeling and neural compression**.
>
> | Target |  PSNR ↑  | SSIM ↑  | LPIPS ↓  | # Params |
> | --- | --- | --- | --- | --- |
> | Raw GS Parameter  | 51.660 | 0.925 | 0.141 | 0.21M |
> | SF Embedding  | 58.619 | 0.980 | 0.043 | 0.20M |
>
> | Target |  PSNR ↑  | SSIM ↑  | LPIPS ↓  | # Params |
> | --- | --- | --- | --- | --- |
> | Raw GS Parameter  | 19.922  | 0.648  | 0.410 | 1.87M |
> | SF Embedding  | 24.395 | 0.804 | 0.261 | 1.85M |
>
> Comparison between Gaussian Neural Fields trained using submanifold field embeddings and raw Gaussian parameters. **Top: ShapeSplat, bottom: Mip-NeRF 360**, see Appendix Fig. 11 for visualization.
>
> ---
>
> **W2**: *“… each latent embedding is learned in isolation without awareness of global scene-level context,this independence could limit the model’s ability to generate … ”*
>
> A:
> We appreciate the reviewer’s comment.
>
> First, if we assume all Gaussian splats in a scene are i.i.d., then encoding each Gaussian primitive independently does hurt spatial/semantic consistency of a scene. However, we point out that **Gaussian splats are highly correlated in a 3D space**. A good embedding on top of explicit space occupancy (i.e., xyz coordinates), can still effectively present some semantic-meaningful information. As we show in the unsupervised clustering experiment (L461 - L469), even with no semantic label provided, Gaussian splats can be clustered well by SF-embedding with cleaner boundary and semantic-meaningful segments (like segmenting legs of tables and table surface). This shows that a good per-Gaussian embedding with coordinates can still work for scene-level tasks, of course, more task-specified learning based on this embedding could potentially be better, which we leave it to future works.
>
> Second, we emphasize that this SF-embedding (based on random data) is data-invariant, and therefore generalizes to any 3DGS data without additional training. This effectively removes a major barrier to downstream adoption, namely the lack of large-scale and general 3DGS datasets, as well as the fact that existing scene-level learning is often domain-dependent. Therefore, it could be possible to utilize the data-invariant property of SF-embedding and learn the embedding-based point cloud model to alleviate the domain-dependent problem.

---

> ### Author Response · Authors · 2025-11-26
> **To Reviewer Jmon (2/2)**
>
> **Q1**: *“… clarify the computational cost of the batched PCA, eigen decomposition, and spherical harmonics fitting …”*
>
> A: We add additional results that resolve this concern in the latest version of our paper, see L518 to L525. Here, we will provide additional analysis of the computational complexity of batched PCA and spherical harmonics fitting.
>
> The PCA + spherical harmonics fitting pipeline is fast, especially when run in batch on a GPU. The key reason is that SH fitting admits a closed-form solution rather than requiring iterative optimization.
>
> In more detail:
>
> - PCA stage: This step (`fit_ellipsoid_pca_torch_batched`) estimates the Gaussian ellipsoid’s center, rotation ($R$), and scales ($(a,b,c)$). The dominant cost is forming the covariance matrices, ($O(B\cdot N)$), which is essentially a batched matrix multiplication (BMM). The subsequent eigen decomposition is on tiny ($3\times 3$) matrices and is therefore almost negligible.
> - SH fitting stage: This step (`fit_sh_torch_batched`) is the crucial one. It is not an iterative procedure like gradient descent and does not refine parameters over many steps. Instead, it is a standard least-squares problem. Given all point directions, we build the SH basis matrix ($B\in\mathbb{R}^{N\times K}$) and solve the normal equations $$x = (B^\top B)^{-1}(B^\top, vals),$$
>
> where ($vals$) are the colors and ($x$) are the desired SH coefficients. In code, this amounts to computing ($B^\top B$) and ($B^\top vals$) with `torch.bmm`, then solving once with `torch.linalg.solve` (or via a Cholesky factorization).
>
> | $P$  | Encode Time | Decode Time (w/o Param. Fitting) | Parameter Fitting Time | GPU Memory |
> | --- | --- | --- | --- | --- |
> | $6 \times 6$ | 0.73s | 1.90s | 0.39s | 1.93GB |
> | $8 \times 8$ | 1.02s | 2.56s | 0.41s | 2.36GB |
> | $12 \times 12$ | 1.72s | 3.72s | 0.48s | 4.32GB |
> | $24 \times 24$ | 6.02s | 12.95s | 0.52s | 15.21GB |
>
> All results in this table show time per 1M Gaussians, tested on RTX 5090 with batch size 4096.

---

### Official Review · Reviewer_WJ16 · 2025-11-01

**Soundness:** 4
**Presentation:** 3
**Contribution:** 4
**Rating:** 8
**Confidence:** 4

**Summary:**

Gaussian splats have been used for 3D scene reconstruction with a lot
of success. Recent works explore mapping the GS parameters to another
embedding space using neural networks to allow applications, such as
generation, editing, compression, and even predicting GS parameters
multi-view image sets directly. Authors here focus on such mappings
and note that the common GS parameterization is not optimal. The main
reasons are non-uniqueness, different parameterization lead to the
radiance fields, and numerical heterogeneity, different parameters
have different magnitudes and such magnitude heterogeneity makes
gradient-descent based learning challenging. They propose another
parameterization as well as an auto-encoder model that maps this
parameterization to an embedding space and back. The representation in
the embedding space is claimed to be more suitable for neural network
training. The model is trained using synthetically generated Gaussians
and applied then to the ShapeSplat and Mip-NeRF 360 datasets. Both
object-level and scene-level experiments are conducted. Comparisons
based on similarity of rasterized images and Gaussian parameters are
presented.

**Strengths:**

+ Authors identify an interesting and important problem.
+ The explanation of the problem is very well done.
+ GS is a very relevant topic and the paper is timely.
+ The proposed approach is sound and based on solid
  theory. Proposition 2 is well appreciated.

**Weaknesses:**

+ The experiments on domain-shifts are not well explained. It is not
  clear what is trained on the datasets and what is tested. I thought
  the proposed method was trained using synthetically generated
  Gaussians. This part needs to be explained better I believe.

**Questions:**

+ The issue of heterogeneity of parameter magnitude is a problem of
  optimization. I wonder whether optimizers that approximates second
  order optimization, such as Adam, would be less susceptible to this
  issue.

---

> ### Author Response · Authors · 2025-11-26
> **To Reviewer WJ16**
>
> We sincerely thank **Reviewer WJ16** for recognizing the importance and clarity of our work. We appreciate your constructive feedback and now address your concerns and questions in detail.
>
> ---
>
> **W1**: *“… the experiments on domain-shifts are not well explained … I thought the proposed method was trained using synthetically generated Gaussians …”*
>
> A: We apologize for the confusion here. We have rephrased the experiment setup in the new version of our paper (L372 to L416). In particular, all the results presented in this experiment (cross-domain reconstruction) are based on real datasets, i.e., training on either Mip-NeRF 360 or ShapeSplat, and testing on the other. The motivation behind the domain shift experiment is to show that the SF-based learning framework enables better generalization with out-of-distribution data, **even without random synthetic dataset**. Moreover, we can also find the embedding trained on synthetic dataset (Table 1) do perform the best, here the domain shift experiment can serve as a baseline.
>
> ---
>
> **Q1**: *“… whether optimizers that approximate second order optimization, such as Adam, would be less susceptible to this issue …”*
>
> A:
> Thanks for the suggestions. First, we would like to note that all experiments in our paper use Adam as the optimizer, we have updated this claim in the new manuscript.
>
> We note that there are other specifically designed techniques in gradient methods that can be used to smooth the parameter distribution of neural networks (like weight decay, gradient clipping). However, setting the learning objective (i.e., embedding) to be smooth and homogeneous provides a more principled approach than applying “patch kits” as this resolves the problem directly. In terms of this, we indicate that the submanifold fields can enable encoder optimization directly on a homogeneous geometry rather than non-Euclidean and heterogeneous parameters.

---

### Official Review · Reviewer_kCGA · 2025-11-02

**Soundness:** 3
**Presentation:** 3
**Contribution:** 3
**Rating:** 8
**Confidence:** 4

**Summary:**

This paper argues that the standard parameter-based representation for 3D Gaussian splatting is not a great fit for neural networks. The limitations in regard to non-uniqueness, numerical inconsistency, and mismatches with learning spaces. This leads to instability, poor generalization, and noise sensitivity. To address this, they propose to model each Gaussian as a continuous field on its iso-probability surface and discretized colored point cloud. This preserves geometry and appearance while ensuring uniqueness and consistency, leading to better learning performance.

**Strengths:**

* The paper convincingly explains why Gaussian parameters are a poor representation for learning. The arguments about non-uniqueness and manifold mismatch are backed by valid theoretical analysis. The contents in sections 3.2 and 3.3 are the major strengths, which are generally convincing.

* The qualitative and quantitative results support the claims about robustness and generalization.

* The paper is generally well written and easy to follow. The graphical illustrations are particularly helpful.

* The unsupervised graph clustering (shown in Figure 5) convincingly communicates how the submanifold field embeddings preserve detailed semantics.

**Weaknesses:**

The ablation study in regard to the proposed design (of the architecture of Figure 2) is missing. Currently, the Behavior/sensitivity studies of Figure 6 are presented as the ablation. The choice radar plot of Figure 6 (c) does not really help. The sensitivity comparison of Figure 4 is unfair because the embedding noise and parametric noise do not mean the same thing. It needs to be either justified in an interpretable way or otherwise must be removed to avoid misleading. The confusion on rotation space claimed with the example of Figure 3 is not very convincing, as the source of the artifacts can be anything. For example, among many arbitrarily oriented splats (from initialization), it could be due to not being able to optimize the opacity parameter. Since there could be many splats (which may effectively not contribute) with undesired rotations, their effect can be minimized by minimizing the opacity only.

**Questions:**

Please, conduct the ablation study, behavior study, and sensitivity study in a structured way.
The ablation study must ablate (remove) the components of the design choices (like the network architecture).
The behavior study must showcase how the method behaves over the variables (like the embedding size, training data).
The sensitivity study must showcase how the input fluctuations impact the output (like the noise study).
Authors are recommended to follow a more formal definition and comment on these modifications during the rebuttal.
Please also address the weaknesses reported above.

---

> ### Author Response · Authors · 2025-11-26
> **To Reviewer kCGA**
>
> We sincerely thank **Reviewer kCGA** for the detailed assessment and now respond to each point in order.
>
> ---
>
> **W1:** *“… the ablation study in regard to the proposed design (of the architecture of Figure 2) is missing …”*
>
> A: We have added an additional ablation study to discuss the design of SF-VAE in L484 - L501. Specifically, we tested a different encoder architecture [1], replaced the unit sphere with 2D grid [2], and attempted a more accurate fitting module. The results support our current design as a good balance between efficiency and performance.
>
> ---
>
> **W2:** *“… the sensitivity comparison of Figure 4 is unfair because the embedding noise and parametric noise do not mean the same thing …”*
>
> A: We apologize for the confusion. We would like to clarify that the noise is not added to the Gaussian parameters directly, rather it is added to the MLP embedding of parameters and the SF embedding for the examination of embedding sensitivity. To ensure fairness of comparison, we actually adjusted the noise level (i.e., ratio to the embedding variance) instead of the absolute noise. We have updated a rephrase of this part in the latest version of our paper(L420 to L426).
>
> ---
>
> **W3:** *“… the confusion on rotation space claimed with the example of Figure 3 is not very convincing … their effect can be minimized by minimizing the opacity only.”*
>
> A: We would like to explain our rationale here.
>
> First, the rotational ambiguity is not an artifact but a consequence of the **embedding failure**. In other words, the encoder has failed to find a stable one-to-one mapping between its rotation-related input/output pairs. In fact, **Appendix Fig. 9** displays this failure: we flip the signs for all rotation quaternions before parameters encoding, which should produce the exact same or similar decoded rotations due to symmetry, but the result is a complete failure. Optimization cannot prevent this failure since it is rooted in the representation itself.
>
> Second, the reason why submanifold field works well with rotation is that it learns rotation in an implicit manner. This learning is much simpler compared to predicting the exact rotation quaternions values directly, since quaternions are discontinuous in Euclidean spaces. This topic is well-explored by Zhou, et al (2019)[3].
>
> ---
>
> **Q1** *"Please, conduct the ablation study, behavior study, and sensitivity study in a structured way."*
>
> Please refer to our response to **W1**, as well as the related section in the latest version of our paper.
>
> ---
>
> [1] Wang, Yue, et al. "Dynamic graph cnn for learning on point clouds." ACM Transactions on Graphics (tog) 38.5 (2019): 1-12.
>
> [2] Yang, Yaoqing, et al. "Foldingnet: Point cloud auto-encoder via deep grid deformation." Proceedings of the IEEE conference on computer vision and pattern recognition. 2018.
>
> [3] Zhou, Yi, et al. "On the continuity of rotation representations in neural networks." Proceedings of the IEEE/CVF conference on computer vision and pattern recognition. 2019.

---

### Author Response · Authors · 2025-11-26
**General Comments and Changelog**

We are grateful for the insightful feedback from all reviewers:
1. Reviewers appreciated the novelty of this work. **Reviewer WJ16** identified *an interesting and important problem,* **Reviewer Jmon** emphasized the representation is *interesting and novel,* and **Reviewer jRea** acknowledged *an interesting and timely idea.*
2. **Reviewers kCGA, WJ16, and Jmon** particularly commended our theoretical contributions. **Reviewer kCGA** highlighted that *The arguments about non-uniqueness and manifold mismatch are backed by valid theoretical analysis.* **Reviewer WJ16** appreciated that *The proposed approach is sound and based on solid theory.*  **Reviewer Jmon** praised that *The paper clearly explains the problem formulation and the motivations.*
3. We also thank the specific commendations: **Reviewer kCGA** commended the unsupervised graph clustering results, **Reviewer WJ16** highlighted Proposition 2, **Reviewer Jmon** valued the learning approach using randomly generated datasets, and **Reviewer jRea** recognized the broad analysis including robustness, clustering, and interpolation evaluations.

To reply the reviewers’ concerns, we have changed/added the following contents in the updated manuscript (with Appendix):

1. New narrative for the problem of raw GS parameters for neural networks (L38-L42)
    - Introduced more literature supports for theoretical evidences.
2. Rephrase the consequences of poor embedding quality in practice (L47-L77)
    - Added a description of the potential consequences in downstream tasks
3. Reorganize Section 4.2 and add a discussion on differences of reconstruction experiments (Section 4.2)
4. Change Table 2 caption, Figure 3 caption, Figure 4 caption and add discussion.
    - Provided more explanation on the experiment settings
5. More results of representation applicability
    - unsupervised clustering task: added more visualizations (Figure 5) and quantitative results (Appendix Table 4)
6. New results of representation applicability
    - Gaussian neural field task (Section 4.4, Figure 6, Table 3)
7. Change Figure 7 (c) for clearness
8. New ablation studies on SF embedding learning framework (SF-VAE) (Section 4.5)
9. Rename behavior study and sensitivity study of embedding (Section 4.3 & Section 4.5)
10. New computational efficiency results (L380)
    - Added decoding time, complexity analysis, parameter fitting breakdown

---

### Author Response · Authors · 2025-12-02
**Summary of Discussion to AC (2/2)**

### The following are the responses summary for each reviewer

**To Reviewer kCGA (rating 8, confidence 4):**

- To address “*ablation study missing*” ([**W1, Q1**](https://openreview.net/forum?id=NvpVtGG6hk&noteId=5CMayz3uCP)) we added ablation studies on the SF-VAE design (encoder choice, decoder grid, fitting module) and updated in the manuscript.
- To address “*sensitivity comparison unfair*” (**W2**), we clarified that the noise in the experiment is variance-normalized and updated the captions.
- To answer “*artifact* *splats with undesired rotations can be minimized*”(**W3**), we explained the pointed “artifacts” to be embedding failures rather than artifacts, which cannot be eliminated entirely if not using our embedding.

**To Reviewer WJ16 (rating 8, confidence 4):**

- To address “*experiments on domain-shifts are not well explained*” **(W1)**, we re-clarified the experimental setup and updated manuscript.
- To answer “*whether second order optimization can be less susceptible to heterogeneity of parameter*” (**Q1**), we discussed that heterogeneity can be addressed better at the representation design inherently than just using optimization techniques.

**To Reviewer Jmon (rating 4, confidence 3):**

- To address “*remains unclear how well this representation generalizes to downstream tasks.*”(**W1**), we added new results on **unsupervised clustering** and **Gaussian neural field** experiments in the manuscript. These successes can be utilized in more downstream tasks.
- To counter “*each latent embedding is learned in isolation, limits generate spatially consistent or semantically coherent context in a scene*” (**W2**), we clarified that although each embedding is learned in instance-level, the associated Gaussian points are *not* i.i.d. Their spatial correlations along with embeddings naturally enforces scene-level consistency, which is showcased in clustering results.
- To “*clarify the computational cost of decoding process”* (**Q1**)**,** **we added new results on high computational efficiency of the PCA + SH fitting module and updated in manuscript.

**To Reviewer jRea (rating 2, confidence 4):**

- To “*elaborate the terms "sub" and "field" in "submanifold fields”* ” (**Q1**) in title, We explained to the reviewer.
- To answer “*is there a specific reason to formalize the non-uniqueness and heterogeneity so strongly?”*(**Q2**), we explained to the reviewer about the meaning of them and updated in manuscript.
- To “*provide the correlation L1 with L2*”, (**Q3**) we clarified that L2 distance works similarly to L1 distance in high-dimensional space for comparison, and indicated that L1 is typically used for optimizing Gaussian Splatting.
- To “*clarify this apparent discrepancy in M-Dist*”, (**Q4**) we apologized about the labeling mistakes and modified in manuscript.
- To address “*include more straightforward or intuitive examples in Figure 5, how the SF-VAE interpolation is superior to the other methods*”,(**Q5**), we indicated that there was Fig. a in Appendix illustrating this interpolation effect. We also explained that jitter in rotation and scale during interpolation suggests that the parametric space is not smooth.
- To address “*… the motivation could be strengthened …by highlighting downstream applications …”* (**W1**)*,* we added a new narrative in Introduction and added more downstream task results in Section 4.4.
- To address*“… experiments focus on small networks … it is unclear whether results will generalize to larger models …”* (**W2**), **we **clarified that size of the network will not affect much towards the encoding a single Gaussian primitive which is shown in Ablation study (Section 4.5)

**To Reviewer jRea New Reply:**

- The reviewer replies on what he/she thinks our work is doing. We believe there is a misunderstanding and attempt to explain:
    - We clarified the scope: our goal is a *learning-task-oriented* representation for 3DGS not to fully replace the raw GS parameters for rendering.
    - We explained the roles of the submanifold field and VAE, and learned embeddings are used as stand-alone inputs in downstream settings (clustering, Gaussian Neural Fields).
    - We highlighted again our theoretical contributions and experimental settings, which aligns well to the purpose of better learning-based tasks on 3DGS data.

---

### Author Response · Authors · 2025-12-02
**Summary of Discussion to AC (1/2)**

We appreciate that all reviewers find our work **theoretically grounded and novel** for representation learning on 3D Gaussian Splatting. Reviewers **kCGA, WJ16,** and **Jmon** highlight our theoretical contribution in formalizing and analyzing Gaussian parameter space, and reviewer **jRea** acknowledges the problem as meaningful and the geometric perspective as interesting. We also thank the reviewer **kCGA**, **WJ16** and **Jmon** for noting the clarity of visualizations.

### Our main contributions are:

- We provide analysis showing that raw 3DGS parameters are a poor learning space due to **non-uniqueness**, **numerical heterogeneity**, and **manifold mismatch**.
- We introduce a submanifold field formulation that associates each Gaussian with a unique and homogeneous geometry, and define the manifold distance based on it.
- We realize this formulation with an designed SF-VAE outputting embeddings. We experimentally show that the embeddings support more **robust, generalizable, and semantically structured** learning than raw parameters across multiple datasets.

### To reply to the reviewers’ concerns, we:

1. **Clarify scope**: we rephrase the introduction and add narrative about **why good GS embeddings are needed** for learning-based tasks while leaving renderer-oriented raw parameters unchanged (Section 1).
2. **More experimental evidence**: we add experimental results on **unsupervised clustering** and **Gaussian Neural Fields**, to show that the proposed embeddings provide **better semantic structure** and are **better regression targets** than raw parameters (Section 4.4, Appendix D.3, D.4).
3. **Analyze computational efficiency**: we add experiments of the decoding pipeline to evaluate its efficiency (Section 4.5).
4. **Add ablations on architecture design**: we add detailed ablations on encoder architecture and implementation of our SF-VAE design (Section 4.5).
5. **Other edits**: other detailed clarifications (figure/table caption changes & reorganizing in Section 4.2–4.4) are summarized in the [**changelog**](https://openreview.net/forum?id=NvpVtGG6hk&noteId=XJIgSnc3PY).

---

### Meta-Review · Area_Chair_VEjQ · 2026-01-06

**Summary:**

The reviewers commend the paper for addressing an important and timely problem in Gaussian Splatting, with a clear and well-motivated formulation grounded in solid theory, particularly appreciating Proposition 2. The idea of learning discretized iso-probability surfaces as a geometric representation for Gaussian primitives is seen as both novel and compelling. The proposed approach is methodologically sound, can be trained in a domain-agnostic manner using randomly generated data, and aligns well with the growing interest in large-scale reconstruction models. Empirically, the method shows strong performance improvements over the baseline and is supported by a broad experimental analysis, including robustness, clustering, and interpolation studies, which together reinforce the paper’s technical merit and relevance.

There were some initial concerns.  These regard some certain comparisons which were potentially misleading, e.g., equating embedding noise with parametric noise but these were were well-addressed in the rebuttal. Finally, the independent encoding of each iso-probability surface without global context have made the reviewers to be concerned about spatial consistency and semantic coherence when scaling to full-scene generation, limiting the practical applicability of the approach.

Overall, the reviewers are fairly positive. There is the situation of reviewer jRea and I've decided to ignore this review. The authors are quite aggressive on this point but I agree that the review quality is rather poor and the authors are right in complaining.

**Reviewer Concerns:**

I think the authors have done a good job with their rebuttal in addressing the received comments. There is one comment that I found valid:  the domain-shift experiments were initially poorly explained, leaving it unclear what data the model is trained and tested on and how synthetic Gaussian training relates to real datasets. More broadly, while the paper motivated iso-probability surfaces as a learning-friendly alternative to parametric Gaussian representations, the experiments largely showed improved parameter reconstruction in controlled settings and did not convincingly demonstrate generalization to downstream or scene-level tasks such as compression or generative modeling. This point has been clearly addressed in the rebuttal so I think the authors have answered it convincingly.

**Reviewer Scores:**

Clearly, the discussion would have given the authors the chance to answer some of the minor comments they received and to hopefully converge with reviewer jRea. I saw the new comment the reviewer posted after reading the rebuttal and I agree that the initial review was completely out of scope. It is hard to immagine the follow-up reaction of the reviewer but clearly the answers of the reviewers have started to make the reviewer understand better the work. There are still some comment from Reviewer Jmon but I think these have been well-addressed in the rebuttal. Overall, I've decided to go with the two positive reviewers which are experts in this field and I trust their judgments.

---

### Decision · Program_Chairs · 2026-01-26

Accept (Poster)